# UniControl: A Unified Diffusion Model for Controllable Visual Generation In the Wild

Can Qin[†*], Shu Zhang[†], Ning Yu[†], Yihao Feng[†], Xinyi Yang[†], Yingbo Zhou[†], Huan Wang[†], Juan Carlos Niebles[†], Caiming Xiong[†], Silvio Savarese[†], Stefano Ermon[‡], Yun Fu[*], and Ran Xu[†]

[†]Salesforce AI Research, [*]Northeastern University, [‡]Stanford Univeristy,
qin.ca@northeastern.edu, ermon@cs.stanford.edu, yunfu@ece.neu.edu,
{shu.zhang, ning.yu, yihaof, x.yang, yingbo.zhou, huan.wang, jniebles,
cxiong, ssavarese, ran.xu}@salesforce.com

## Abstract

Achieving machine autonomy and human control often represent divergent objectives in the design of interactive AI systems. Visual generative foundation models such as Stable Diffusion show promise in navigating these goals, especially when prompted with arbitrary languages. However, they often fall short in generating images with spatial, structural, or geometric controls. The integration of such controls, which can accommodate various visual conditions in a single unified model, remains an unaddressed challenge. In response, we introduce UniControl , a new generative foundation model that consolidates a wide array of controllable condition-to-image (C2I) tasks within a singular framework, while still allowing for arbitrary language prompts. UniControl enables pixel-level-precise image generation, where visual conditions primarily influence the generated structures and language prompts guide the style and context. To equip UniControl with the capacity to handle diverse visual conditions, we augment pretrained text-to-image diffusion models and introduce a task-aware HyperNet to modulate the diffusion models, enabling the adaptation to different C2I tasks simultaneously. Trained on nine unique C2I tasks, UniControl demonstrates impressive zero-shot generation abilities with unseen visual conditions. Experimental results show that UniControl often surpasses the performance of single-task-controlled methods of comparable model sizes. This control versatility positions UniControl as a significant advancement in the realm of controllable visual generation. [1]

## 1 Introduction

Generative foundation models are revolutionizing the ways that humans and AI interact in natural language processing (NLP) [1–6], computer vision (CV) [7–10], audio processing (AP) [11, 12], and robotic controls [13–15], to name a few. In NLP, generative foundation models such as InstructGPT or GPT-4, achieve excellent performance on a wide range of tasks, *e.g.,* question answering, summarization, text generation, or machine translation within a *single-unified* model. Such multi-tasking ability is one of the most appealing characteristics of generative foundation models. Furthermore, generative foundation models can also perform zero-shot or few-shot learning on unseen tasks [3, 16, 17].

For generative models in vision domains [9, 18–20], such multi-tasking ability is less clear. Stable Diffusion Model (SDM) [9] has established itself as the major cornerstone for text-conditioned image generation. However, while text descriptions provide a very flexible way to control the generated images, their ability to provide pixel-level precision for spatial, structural, or geometric controls is

---

[1]Code: https://github.com/salesforce/UniControl

37th Conference on Neural Information Processing Systems (NeurIPS 2023).

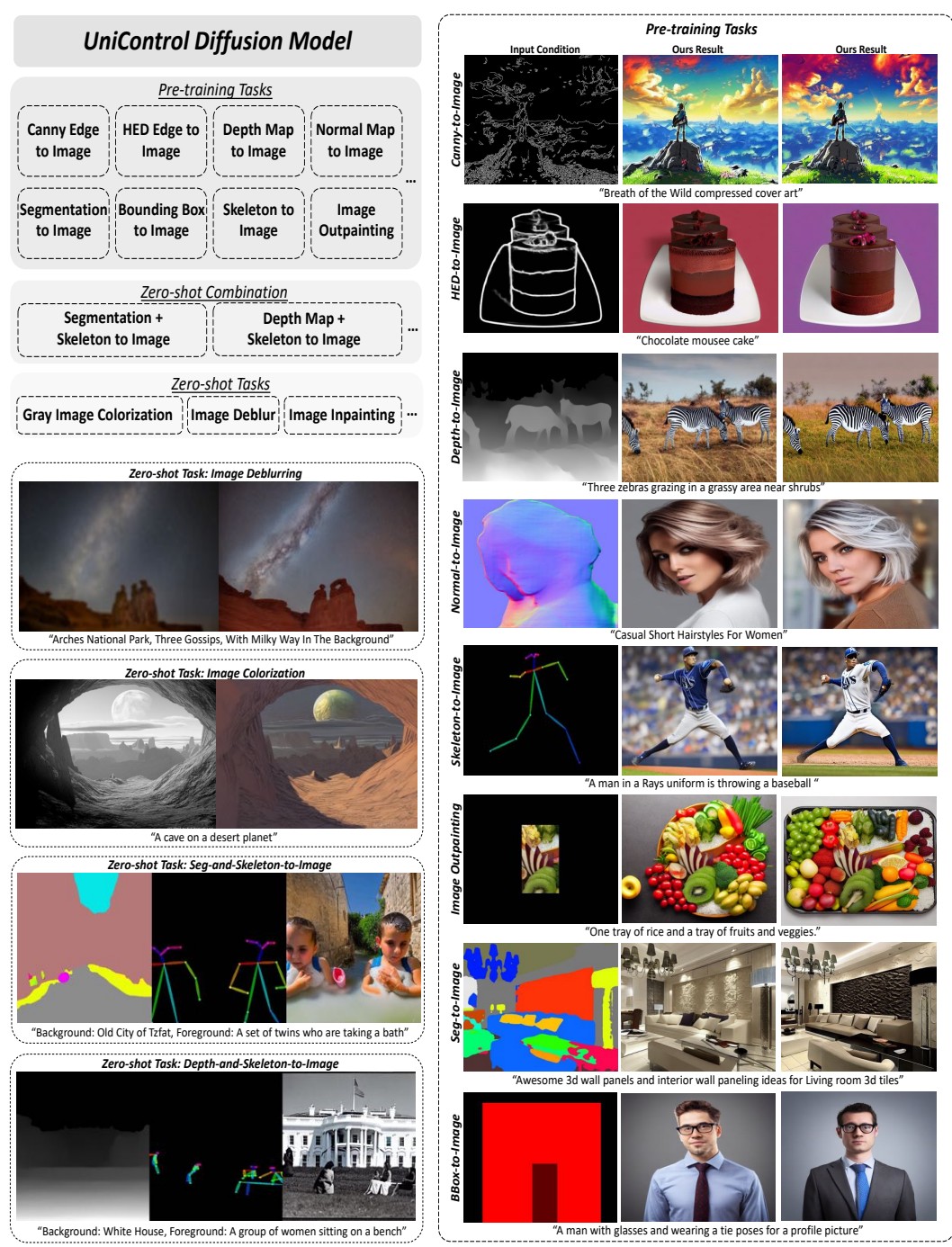

Figure 1: UniControl is trained with multiple tasks with a unified model, and it further demonstrates promising capability in zero-shot tasks generalization with visual example results shown above.

often inadequate. A recent work, ControlNet [21], was proposed to augment SDM to enable visual conditions (*e.g.,* edge maps, depth maps). With the additional visual conditions, ControlNet can achieve explicit spatial, structural, or geometric control over generated structures, without losing the semantic control from textual captions. Unfortunately, unlike language prompts that a unified module such as CLIP [22] can handle, each ControlNet model can only handle a specific control modality that it was trained on (*e.g.*, edge map). Retraining a separate model is necessary to handle a different modality of visual conditions, incurring non-trivial time and spatial complexity costs.

To overcome the limitation of previous works, we present **UniControl**, a unified diffusion model for controllable visual generation in the wild, which is capable of simultaneously handling both

language and various visual conditions. Naturally, UniControl can perform multi-tasking and can encode visual conditions from different tasks into a universal representation space, seeking a common representation structure among tasks. The unified design of UniControl allows us to enjoy the advantages of improved training and inference efficiency, as well as enhanced controllable generation. On the one hand, the model size of UniControl does not significantly increase as the number of tasks scales up. On the other hand, UniControl derives advantages from the inherent connections between different visual conditions [*e.g.*, 23–25]. These relationships, such as depth and segmentation mapping, leverage shared geometric information to enhance the controllable generation quality.

The unified controllable generation ability of UniControl relies on two novel designed modules, a *mixture of expert (MOE)-style adapter* and a *task-aware HyperNet* [26, 27]. The MOE-style adapter can learn necessary low-level feature maps from various visual conditions, allowing UniControl to capture unique information from different visual conditions. The task-aware HyperNet, which takes the task instruction as natural language prompt inputs, and outputs a task-aware embedding. The output embeddings can be incorporated to modulate ControlNet [21] for task-aware visual condition controls, where each task corresponds to a particular format of visual condition. As a result, the task-aware HyperNet allows UniControl to learn meta-knowledge across various tasks, and obtain abilities to generalize to unseen tasks. As Tab. 1, UniControl has significantly compressed the model size compared with its direct baseline, *i.e.*, Multi-ControlNet, by unifying nine tasks into **ONE** model.

Table 1: Architecture and Model Size (#Params): UniControl *vs.* Multi-ControlNet

|  | Stable Diffusion | ControlNet | MoE-Adapter | TaskHyperNet | Total |
|---|---|---|---|---|---|
| **UniControl** | 1065.7M | 361M | 0.06M | 12.7M | **1.44B** |
| **Multi-ControlNet** | 1065.7M | 361M × 9 | - | - | 4.32B |

To obtain multi-tasking and zero-shot learning abilities, we pre-train UniControl on nine distinct tasks across five categories: **1)** *edges* (Canny, HED, User Sketch); **2)** *region-wise maps* (Segmentation Maps, Bounding Boxes); **3)** *skeletons* (Human Pose Skeletons); **4)** *geometric maps* Depth, Surface Normal); **5)** *editing* (Image Outpainting). We build **MultiGen-20M** dataset, comprising over 20 million high-quality triplets of original images, language prompts, and visual conditions for all the tasks. Then UniControl is trained for over 5,000 GPU hours on NVIDIA A100-40G hardware that is comparable with the overall training cost of different ControlNets. Moreover, UniControl exhibits a remarkable capacity for zero-shot adaptation to new tasks, highlighting its potential for deployment in real-world applications. Our contributions are summarized below:

- We present UniControl, a unified model capable of handling various visual conditions for the controllable visual generation.

- We collect a new dataset for multi-condition visual generation with more than 20 million image-text-condition triplets over nine distinct tasks across five categories.

- We conduct extensive experiments to demonstrate that the unified model UniControl outperforms each single-task controlled image generation, thanks to learning the intrinsic relationships between different visual conditions.

- UniControl shows the ability to adapt to unseen tasks in a zero-shot manner, highlighting its versatility and potential for widespread adoption in the wild.

## 2 Related Works

**Diffusion-based Generative Models.** Diffusion models were initially introduced in [28] that yield favorable outcomes for generating images [18, 21]. Improvements have been made through various training and sampling techniques such as score-based diffusion [29, 30], Denoising Diffusion Probabilistic Model (DDPM) [31], and Denoising Diffusion Implicit Model (DDIM) [32], When training U-Net denoisers [33] with high-resolution images, researchers involve speed-up techniques including pyramids [34], multiple stages [20], or latent representations [9]. In particular, UniControl leverages Stable Diffusion Models (SDM) [9] as the base model to perform multi-tasking.

**Text-to-Image Diffusion.** Diffusion models emerge to set up a cutting-edge performance in text-to-image generation tasks [20, 19], by cross-attending U-Net denoiser in diffusion generators with CLIP [22] or T5-pretrained [2] text embeddings. GLIDE [35] is another example of a text-guided diffusion model that supports image generation and editing. UniControl and closely related

ControlNet [21] are both built upon previous works on diffusion-based text-to-image generation [9]. [36] introduces the compositional conditions to guide visual generation.

**Image-to-Image Translation.** Image-to-image (I2I) translation task was initially proposed in Pix2Pix [37], focusing on learning a mapping between images in different domains. Recently, diffusion-based approaches [38, 39, 21] set up the new state of the art results. Recent diffusion-based image editing methods show outstanding performances without requiring paired data, *e.g.,* SDEdit [40], prompt-to-prompt [41], Edict [42]. Other image editing examples include various diffusion bridges and flows [43–47], classifier guidance [30] based methods for colorization, super-resolution [34], inpainting [48], and *etc*. ControlNet [21] takes both visual and text conditions and achieves new state-of-the-art controllable image generation. Our proposed UniControl unifies various visual conditions of ControlNet, and is capable of performing zero-shot learning on newly unseen tasks. Concurrently, Prompt Diffusion [49] introduces visual prompt [50] from image inpainting to controllable diffusion models, which requires two additional image pairs as the in-context example for both training and inference. By contrast, UniControl takes only a single visual condition while still capable of both multi-tasking and zero-shot learning.

## 3 UniControl

In this section, we describe the training and the model design of our unified controllable diffusion model **UniControl**. Specifically, we first provide the problem setup and training objectives in Sec. 3.1, and then show the novel network design of UniControl in Sec. 3.2. Finally, we explain how to perform zero-shot image generation with the trained UniControl in Sec. 3.3.

### 3.1 Training Setup

Different from the previous generative models such as Stable Diffusion Models (SDM) [9] or ControlNet [21], where the image generation conditions are *single* language prompt, or *single* type of visual condition such as *canny*, UniControl is required to take a wide range of visual conditions from different tasks, as well as the language prompt.

To achieve this, we reformulate the training conditions and target pairs for UniControl. Specifically, suppose we have a dataset consisting of $K$ tasks : $\mathcal{D} := \{\mathcal{D}_1 \cup \cdots \cup \mathcal{D}_K\}$, and for each task training set $\mathcal{D}_k$, denote the training pairs by $([c_{\text{text}}, c_{\text{task}}], \mathcal{I}_c, \boldsymbol{x})$, with $c_{\text{task}}$ being the task instruction that indicates the task type, $c_{\text{text}}$ being the language prompt describing the target image, $\mathcal{I}_c$ being the visual conditions, and $\boldsymbol{x}$ being the target image. With the additional task instruction, UniControl can differentiate visual conditions from different tasks. A concrete training example pair is the following:

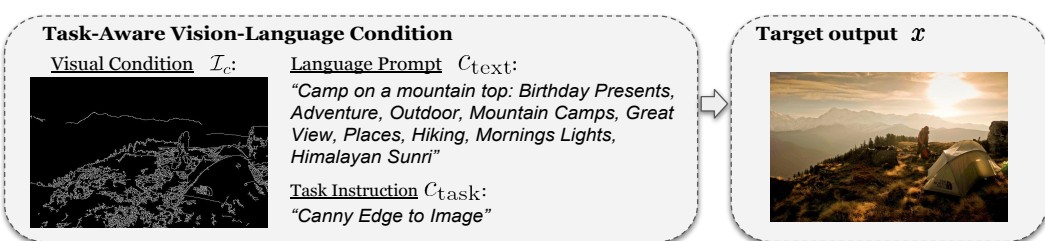

where the task is to *translate the canny edge to real images following language prompt.* With the induced training pairs $(\boldsymbol{x}, [c_{\text{task}}, c_{\text{text}}], \mathcal{I}_c)$, we define the training loss for task $k$ following LDM [9]:

$$\ell^k(\theta) := \mathbb{E}_{z, \varepsilon, t, c_{\text{task}}, c_{\text{text}}, \mathcal{I}_c} \left[ \|\varepsilon - \varepsilon_\theta(z_t, t, c_{\text{task}}, c_{\text{text}}, \mathcal{I}_c)\|_2^2 \right], \text{ with } ([c_{\text{task}}, c_{\text{text}}], \mathcal{I}_c, \boldsymbol{x}) \sim \mathcal{D}_k,$$

where $t$ represents the time step, $z_t$ is the noise-corrupted latent tensor at time step $t$, $z_0 = E(\boldsymbol{x})$, and $\theta$ is the trainable parameters of UniControl . We also apply classifier-free guidance [51] to randomly drop 30% text prompts to enhance the controllability of input visual conditions. We train UniControl uniformly on the $K$ tasks. To be more specific, we first randomly select a task $k$ and sample a mini-match from $\mathcal{D}_k$, and optimize $\theta$ with the calculated loss $\ell^k(\theta)$.

### 3.2 Model Design

Since our unified model UniControl needs to achieve superior performance on a set of diverse tasks, it is necessary to ensure the network design enjoys the following properties: **1)** The model can overcome

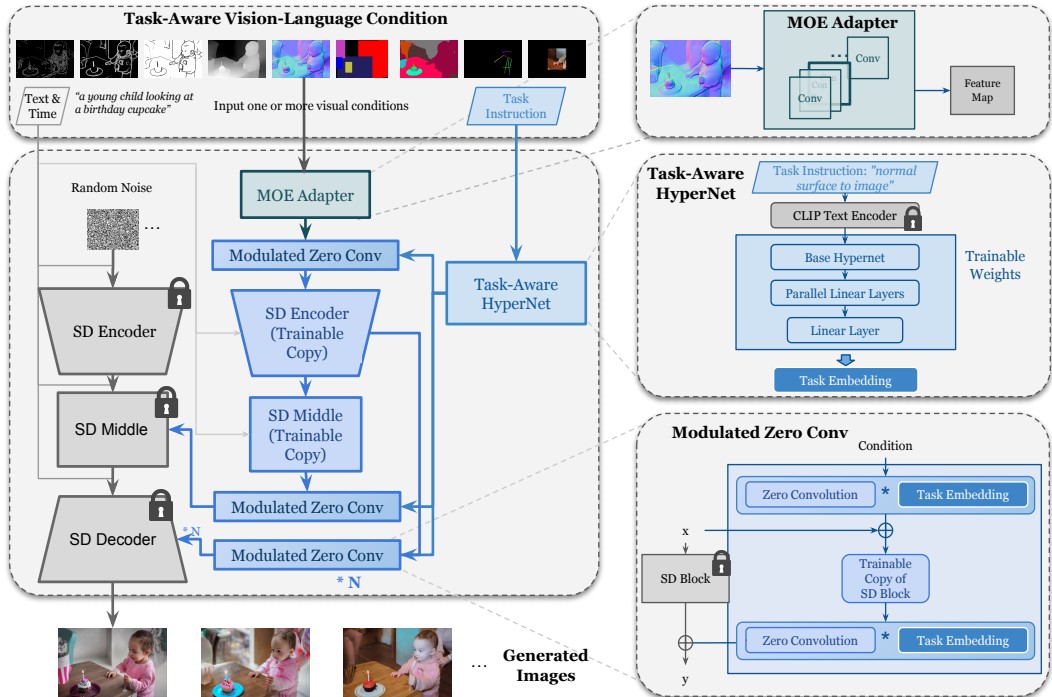

Figure 2: This figure shows our proposed UniControl method. To accommodate diverse tasks, we've designed a Mixture of Experts (MOE) Adapter, containing roughly ∼70K #params for each task, and a Task-aware HyperNet (∼12M #params) to modulate $N$ (i.e., 7) zero-conv layers. This structure allows for multi-task functionality within a singular model, significantly reducing the model size compared to an equivalent stack of single-task models, each with around 1.4B #params.

the misalignment of low-level features from different tasks; **2)** The model can learn meta-knowledge across tasks, and adapt to each task effectively.

The first property can ensure that UniControl can learn necessary and unique information from all tasks. For instance, if UniControl takes the segmentation map as the visual condition, the model might ignore the 3D information. As a result, the feature map learned may not be suitable for the task that takes the depth map images as visual condition. The second property would allow the model to learn the shared knowledge across tasks, as well as the differences among them.

We introduce two novel designed modules, *MOE-style adapter* and *task-aware HyperNet*, that allows UniControl enjoys the above two properties. An overview of the model design for UniControl is in Fig. 2. We describe the detailed designs of these modules below.

**MOE-Style Adapter.**   Inspired by the design of Mixture-of-Experts (MOEs) [52], we devise a group of convolution modules to serve as the adapter for UniControl to capture features of various low-level visual conditions. Precisely, the designed adapter module can be expressed as

$$\mathcal{F}_{\text{Adapter}}(\mathcal{I}_c^k) := \sum_{i=1}^{K} \mathbb{1}(i == k) \cdot \mathcal{F}_{\text{Cov1}}^{(i)} \circ \mathcal{F}_{\text{Cov2}}^{(i)}(\mathcal{I}_c^k),$$

where $\mathbb{1}(\cdot)$ is the indicator function, $\mathcal{I}_c^k$ is the conditioned image from task $k$, and $\mathcal{F}_{\text{Cov1}}^{(i)}, \mathcal{F}_{\text{Cov2}}^{(i)}$ are the convolution layers of the $i$-th module of the adapter. We remove the weights of the original MOEs since our designed adapter is required to differentiate various visual conditions. Meanwhile, naive MOE modules can not explicitly distinguish different visual conditions when the weights are learnable. Moreover, such task-specific MOE adapters facilitate the zero-shot tasks with explicit retrieval of the adapters of highly related pre-training tasks. Besides, the number of parameters for each convolution module is approximately 70K, which is computationally efficient.

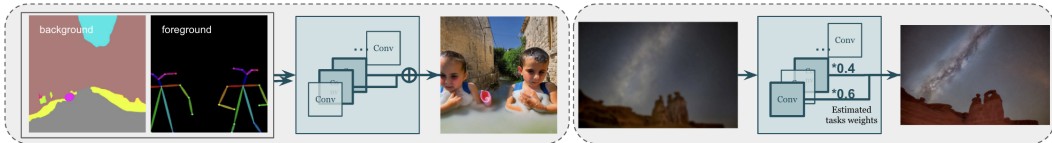

| MOE Hybrid Tasks Generalization | MOE Zero-Shot New Task Generalization |

Figure 3: Illustration of MOE's behaviors under zero-shot scenarios. The left part shows the capacity of the MOE to generalize to hybrid task conditions, achieved through the integration of outputs from two pertinent convolution layers. The right part illustrates the ability of the MOE-style adapter to generalize to unseen tasks, facilitated by the aggregation of pre-trained tasks using estimated weights.

**Task-Aware HyperNet.** The task-aware HyperNet modulates the zero-convolution modules of ControlNet [21] with the task instruction condition $c_{task}$. As shown in Figure 2, our hyperNet first projects the task instruction $c_{task}$ into task embedding with the help of CLIPText encoder. Then similar in spirit of style modulation in StyleGAN2 [53], we inject the task embedding into the trainable copy of ControlNet, by multiplying the task embedding to each zero-conv layer. In specific, the length of the embedding is the same as the number of input channels of the zero-conv layer, and each element scalar in the embedding is multiplied to the convolution kernel per input channel. We also show that our newly designed task-aware HyperNet can also efficiently learn from training instances and task supervision following a similar analysis as in ControlNet [21].

### 3.3 Task Generalization Ability

With the comprehensive pretraining on the MultiGen-20M dataset, UniControl exhibits zero-shot capabilities on tasks that were not encountered during its training, suggesting that Unicontrol possesses the ability to transcend in-domain distributions for broader generalization. We demonstrate the zero-shot ability of UniControl in the following two scenarios:

**Hybrid Tasks Generalization.** As shown in the left side of Fig. 3, We consider two different visual conditions as the input of UniControl, a hybrid combination of segmentation maps and human skeletons, and augment specific keywords "background" and "foreground" into the text prompts. Besides, we rewrite the hybrid task instruction as a blend of instructions of the combined two tasks such as "segmentation map and human skeleton to image".

**Zero-Shot New Tasks Generalization.** As shown in the right side of Fig. 3, UniControl needs to generate controllable images on a newly unseen visual condition. To achieve this, estimating the task weights based on the relationship between unseen and seen pre-trained tasks is essential. The task weights can be estimated by either manual assignment or calculating the similarity score of task instructions in the embedding space. The example result in Fig. 5 (d) is generated by our manually assigned MOE weights as "depth: 0.6, seg: 0.3, canny: 0.1" for colorization. The MOE-style adapter can be linearly assembled with the estimated task weights to extract shallow features from the newly unseen visual condition.

## 4 Experiments

We empirically evaluate the effectiveness and robustness of UniControl. We conduct a series of comprehensive experiments across various conditions and tasks, utilizing diverse datasets to challenge the model's adaptability and versatility. Experimental setup, methodologies, and results analysis are provided in the subsequent sections.

### 4.1 Experiment Setup

**Implementation.** The UniControl is illustrated as Fig. 2 with Stable Diffusion, ControlNet, MOE Adapter, and Task-aware HyperNet consisting ~1.5B parameters. MOE Adapter consists of parallel convolutional modules, each of which corresponds to one task. The task-aware HyperNet inputs the CLIP text embedding [22] of task instructions and outputs the task embeddings to modulate the weights of zero-conv kernels. We implement our model upon the ControlNet . We take the AdamW [54] as the optimizer based on PyTorch Lightning [55]. The learning rate is assigned as $1 \times 10^{-5}$. Our full-version UniControl model is trained on 16 Nvidia-A100 GPUs with the batch size of 4, requiring $\sim 5,000$ GPU hours. We have also applied Safety-Checker as safeguards of results.

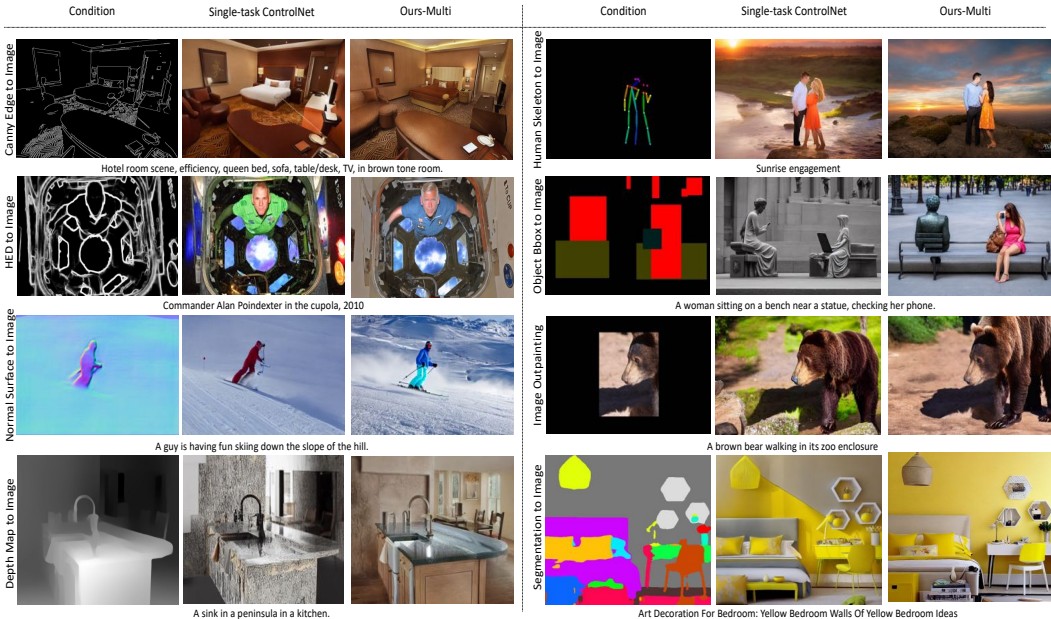

Figure 4: Visual comparison between official or re-implemented task-specific ControlNet and our proposed model. The example data is collected from our testing set sampled from COCO and Laion.

**Data Collection.** Since the training set of ControlNet is currently unavailable, we initiate our own data collection process from scratch and name it as **MultiGen-20M**. We use a subset of Laion-Aesthetics-V2 [56] with aesthetics ratings over six, excluding low-resolution images smaller than 512. This yields approximately 2.8 million image-text pairs. Subsequently, we process this dataset for nine distinct tasks across five categories (edges, regions, skeletons, geometric maps, real images):

- **Canny (2.8M)**: Utilize the Canny edge detector [57] with randomized thresholds.

- **HED (2.8M)**: Deploy the Holistically-nested edge detection [58] for robust boundary determination.

- **Depth (2.8M)**: Employ the Midas [59] for monocular depth estimation.

- **Normal (2.8M)**: Use the depth estimation results from the depth task to estimate scene or object surface normals.

- **Segmentation (2.8M)**: Implement the Uniformer [60] model, pre-trained on the ADE20K [61] dataset, to generate segmentation maps across 150 classes.

- **Object Bounding Box (874K)**: Utilize YOLO V4 [62] pre-trained on the COCO [63] dataset for bounding box labelling across 80 object classes.

- **Human Skeleton (1.3M)**: Employ the pre-trained Openpose [64] model to generate human skeleton labels from source images.

- **Image Outpainting (2.8M)**: Create boundary masks for source images with random masking percentages from 20% to 80%.

Further processings are carried out on HED maps using Gaussian filtering and binary thresholding to simulate user sketching. Overall, we amass over 20 million image-prompt-condition triplets. Task instructions were naturally derived from the respective conditions, with each task corresponding to a specific instruction, such as "canny edge to image" for the canny task. We maintain a one-to-one correspondence between tasks and instructions without introducing variance to ensure stability during training. We have additionally collected a testing dataset for evaluation with 100-300 image-condition-prompt triplets for each task. The source data is collected from Laion and COCO. **We will open-source our training and testing data to contribute to the community.**

**Benchmark Models.** The most straightforward comparison for UniControl comes from task-specific ControlNet models. Six tasks overlap with those presented in ControlNet, so their official models are chosen as baselines for these tasks. For fair comparison, we re-implement the ControlNet model (single task) using our collected data. Our unified multi-task UniControl is compared against these

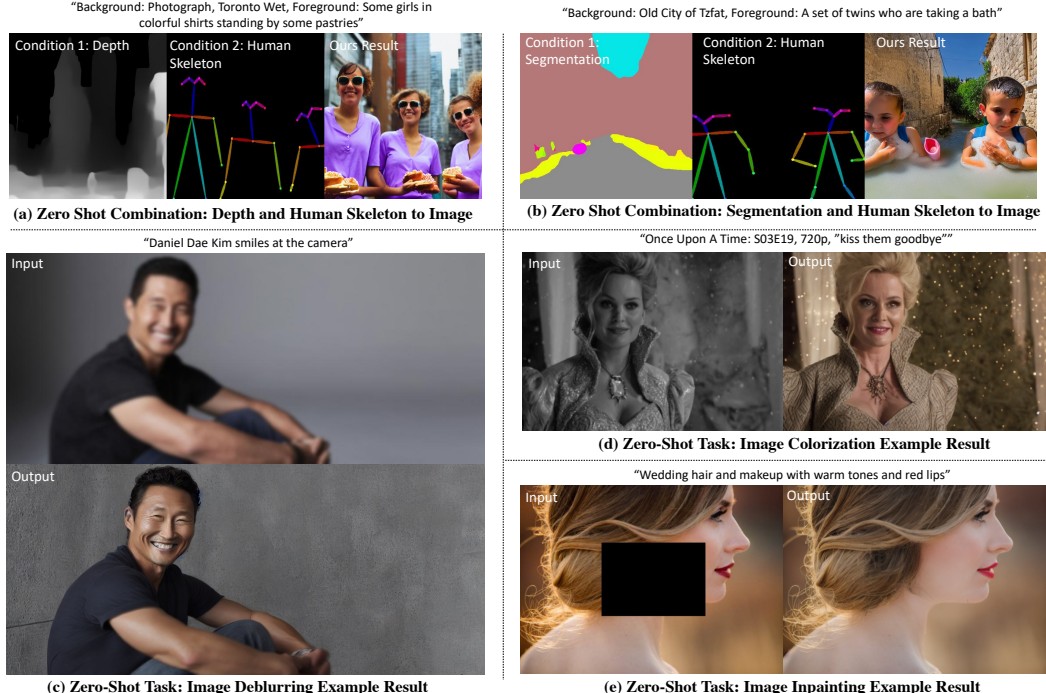

Figure 5: (a)-(b): Example results of UniControl over hybrid (unseen combination) conditions with key words "background" and "foreground" attached in prompts. (c)-(e): Example results of UniControl on three unseen tasks (deblurring, colorization, inpainting).

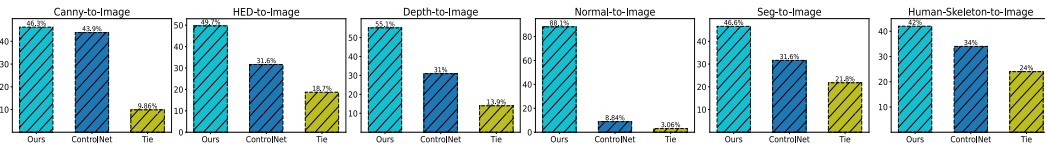

Figure 6: User study between our method and official ControlNet checkpoints on six tasks. Our method outperforms ControlNet on all tasks.

task-aware models for each task. We apply default sampler as DDIM [32] with guidance weight 9 and steps 50. All single-task models used for comparison are trained by 100K iterations and our multi-task model is trained around 900K with similar iterations for each task to ensure fairness. The efficiency and compact design of our proposed model are evident in its construction. The total size of UniControl is around 1.5B #params and a single task ControlNet+SDM takes 1.4B. In order to achieve the same nine-task functionality, a single-task strategy would require the ensemble of a SDM with nine task-specific ControlNet models, amounting to approximately 4.3B #params in total.

## 4.2 Visual Comparison

We visually compare different tasks (Canny, HED, Depth, Normal, Segmentation, Openpose, Bounding Box, and Outpainting) in Fig. 4. Our method consistently outperforms the baseline ControlNet model. This superiority is in terms of both visual quality and alignment with conditions or prompts.

For the Canny task, the results generated by our model exhibit a higher degree of detail preservation and visual consistency. The outputs of UniControl maintain a faithful reproduction of the edge information (*i.e.,* round table) compared to ControlNet. In the HED task, our model effectively captures the robust boundaries, leading to visually appealing images with clear and sharp edge transitions, whereas ControlNet results appear to be non-factual. Moreover, our model demonstrate a more subtle understanding of 3D geometrical guidance of depth maps and surface normals than ControlNet. The depth map conditions produce visibly more accurate outputs. In the Normal task, our model faithfully reproduces the normal surface information (*i.e.,* ski pole), leading to more realistic

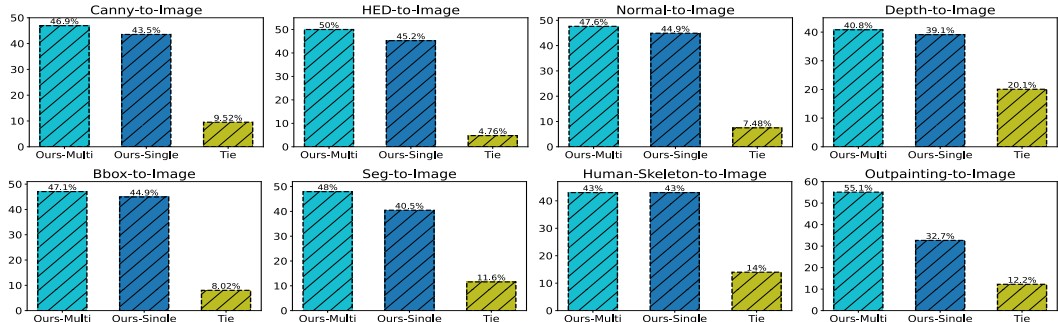

Figure 7: User study between our multi-task model (Ours-multi) and single task model (Ours-single) on eight tasks. Our method outperforms baselines on most of tasks, and achieves big performance gains on tasks of seg-to-image and outpainting-to-image. Moreover, the p-value of voting Ours-multi in all cases is computed as **0.0028** that is statistically significant according to the criteria of <0.05.

and visually superior outputs. During the Segmentation, Openpose, and Object Bounding Box tasks, the produced images generated by our model are better aligned with the given conditions than that by ControlNet, ensuring a higher fidelity to the input prompts. For example, the re-implemented ControlNet-BBox misunderstands "a woman near a statue", whereas our outputs exhibit a high degree of accuracy and detail. In the Outpainting task, our model demonstrates its superiority by generating reasonable images with smooth transitions and natural-looking textures. It outperforms the ControlNet model, which produces less coherent results - "a bear missing one leg". This visual comparison underscores the strength and versatility of our approach across a diverse set of tasks.

### 4.3 Quantitative Evaluation

**User Study.** We compare the performance of our method with both the released ControlNet model and the re-implemented single-task ControlNet on our training set. As shown in Fig. 6, our approach consistently outperforms the alternatives in all cases. In the HED-to-image generation task, our method significantly surpasses ControlNet. This superiority is even more pronounced in the depth and normal surface to image generation tasks, where users overwhelmingly favor our method, demonstrating its ability to handle complex geometric interpretations. When compared to the re-implemented single-task model, Fig. 7 reveals that our approach maintains a smaller advantage, yet it still demonstrates its benefits by effectively discerning image regions to guide content generation. Even in the challenging outpainting task, our model outperforms the baseline, highlighting its robustness and capacity to generalize.

**Image Perceptual Metric.** We evaluate the distance between our output and the ground truth image. As we aim to obtain

Table 2: Image Perceptual Distance

| | Canny ↓ | HED ↓ | Normal ↓ | Depth ↓ | Pose ↓ | Segmentation ↓ |
|---|---|---|---|---|---|---|
| **UniControl** | **0.546** | **0.466** | **0.623** | **0.654** | **0.741** | **0.693** |
| **ControlNet** | 0.577 | 0.582 | 0.778 | 0.700 | 0.747 | **0.693** |

a structural similar image to the ground truth image, we adopt the perceptual metric in [65], where a lower value indicates more similar images. As shown in Tab. 2, UniControl outperforms ControlNet on five tasks, and obtains the same image distance to ControlNet on Segmentation.

**Fréchet Inception Distance (FID).** We've further conducted quantitative analysis with FID [66] to include more classic single-task-controlled methods such as GLIGEN [67] and T2I-adapter [68]. With a collection of over 2,000 test samples sourced from Laion and COCO, we've assessed a wide range of tasks covering edges (Canny, HED), regions (Seg), skeletons (Pose), and geometric maps (Depth, Normal). The Tab. 3 demonstrates that our UniControl consistently surpasses the baseline methods across the majority of tasks. Notably, UniControl achieves this while maintaining a more compact and efficient architecture than its counterparts.

**Ablation Study.** We've conducted an ablation study, specifically focusing on the MoE-Style Adapter and TaskHyperNet in Tab. 4 with FID scores reported as the previous part. It is noticeable that the full-version UniControl (MoE-Style Adapter + TaskHyperNet) significantly outperforms the ablations which demonstrates the superiority of proposed MoE-Style Adapter and TaskHyperNet.

Table 3: Quantitative Comparison (FID)

|  | Canny ↓ | HED ↓ | Depth ↓ | Normal ↓ | Seg ↓ | Pose ↓ |
|---|---|---|---|---|---|---|
| **GLIGEN** [67] | 24.9 | 27.8 | 25.8 | 27.7 | - | - |
| **T2I-Adapter** [68] | 23.6 | - | 25.4 | - | 27.1 | 28.9 |
| **ControlNet** [21] | **22.7** | 25.1 | 25.5 | 28.4 | 26.7 | 28.8 |
| **UniControl** | 22.9 | **23.6** | **21.3** | **23.4** | **25.5** | **27.4** |

Table 4: Ablation Study (FID)

| MoE-Adapter | TaskHyperNet | Canny ↓ | HED ↓ | Depth ↓ | Normal ↓ | Seg ↓ | Pose ↓ | Avg ↓ |
|---|---|---|---|---|---|---|---|---|
| ✗ | ✗ | 27.2 | 29.0 | 27.6 | 28.8 | 29.1 | 30.2 | 28.7 |
| ✓ | ✗ | 24.5 | 26.1 | 23.7 | 24.8 | 26.9 | 28.3 | 25.7 |
| ✓ | ✓ | **22.9** | **23.6** | **21.3** | **23.4** | **25.5** | **27.4** | **24.0** |

## 4.4 Zero-shot Generalization

We further showcase the surprising capabilities of our method to undertake the zero-shot challenge of hybrid conditions combination and unseen tasks generalization.

**Hybrid Tasks Combination.** This involves generating results from two distinct conditions simultaneously. Our model's zero-shot ability is tested with combinations such as depth and human skeleton or segmentation map and human skeleton. The results are shown in Fig. 5 (a)-(b). When the background is conditioned on a depth map, the model effectively portrays the intricate 3D structure of the scene, while maintaining the skeletal structure of the human subject. Similarly, when the model is presented with a combination of a segmentation map and human skeleton, the output skillfully retains the structural details of the subject, while adhering to the segmentation boundaries. These examples illustrate our model's adaptability and robustness, highlighting its ability to handle complex hybrid tasks without any prior explicit training.

**Unseen Tasks Generalization.** To evaluate the zero-shot ability to generalize to unseen tasks such as gray image colorization, image deblurring, and image inpainting, we conduct the case analysis in Fig. 5 (c)-(e). The model skillfully handles the unseen tasks, producing compelling results. This capability is deeply rooted in the shared attributes and implicit correlations among pre-training and new tasks, allowing our model to adapt seamlessly. For instance, the colorization task leverages the model's understanding of image structures from the segmentation task and depth estimation task, while deblurring and inpainting tasks benefit from the model's familiarity with edge detection and outpainting ones.

## 5 Conclusion and Discussion

We introduce UniControl , a novel unified model for incorporating a wide range of conditions into the generation process of diffusion models. UniControl has been designed to be adaptable to various tasks through the employment of two key components: a Mixture-of-Experts (MOE) style adapter and a task-aware HyperNet. The experimental results have showcased the model's robust performance and adaptability across different tasks and conditions, demonstrating its potential for handling complex text-to-image generation tasks.

**Limitation and Broader Impact.** While UniControl demonstrates impressive performance, it still inherits the limitation of diffusion-based image generation models. Specifically, it is limited by our training data, which is obtained from a subset of the Laion-Aesthetics datasets. We observe that there is a data bias in this dataset. Although we have performed keywords and image based data filtering methods, we are aware that the model may generate biased or low-fidelity output. Our model is also limited when high-quality human output is desired. UniControl could be improved if better open-source datasets are available to block the creation of biased, toxic, sexualized, or other harmful content. We hope our work can motivate researchers to develop visual generative foundation models.

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

# Appendix

## A  Details of Implementation

### A.1  MOE-Style Adapter

The MOE adapter is implemented as a set of parallel ConvNets composed of three consecutive convolution and non-linear activation layers. The entire model is comprised of nine individual MOE adapters, each of which consumes 70K parameters. Task keys are designated to each adapter, ensuring that they align with the corresponding visual conditions. Once the MOE adapter processes the input, the remaining model parameters become shared across all tasks. This architecture facilitates task adaptability while promoting parameter efficiency.

### A.2  Task-aware HyperNet

The task-aware hypernet is applied to modulate the parameters of zero-conv layers in the ControlNet. Since the ControlNet can be considered as the hypernet of Stable Diffusion (fixed copy). Our idea can be concluded as the **control over control** or **meta-control** to let the task-aware hypernet learn the universe representation that is generalizable across different tasks. To implement it, we firstly map the task keys to instruction with a mapping function as: `{"hed": "hed edge to image", "canny": "canny edge to image", "seg": "segmentation map to image", "depth": "depth map to image", "normal": "normal surface map to image", "pose": "human pose skeleton to image", "hedsketch": "sketch to image", "bbox": "bounding box to image", "outpainting": "image outpainting"}`. Then, such instructions will be projected as text embeddings with the help of a language model (we adopt CLIPText in our implementation). The Task-aware HyperNet takes these task instruction embeddings, and projects them into different shapes to match the size of different zero-conv kernels, which will be modulated by these task embeddings accordingly. We would fix the parameters of task-aware hyperNet in the later stage of model training to ensure the stability of dynamics.

### A.3  Data Collection

We have collected a large amount of training set (MultiGen-20M) including over 20M condition-image-prompt triplets across nine different tasks. We firstly download 3/4 of Laion-Aesthetics-V2 with score over six and filter out low-resolution (<512) images. As a result, 2.8M images are selected as source images. Then we apply the visual condition extractors as described in the main paper to collect `Canny`, `HED`, `Sketch`, `Depth`, `Normal Surface`, `Seg Map`, `Object Bounding Box`, `Human Skeleton` and `Outpainting`.

## B  Numerical Analysis of Task-Aware Modulated ControlNet

We show that our proposed task-aware modulated ControlNet preserves the properties of the original ControlNet structure. Specifically, we show **1)** The new task-aware modulated ControlNet preserves the zero-initialization property of ControlNet; **2)** The parameters of the task-aware modulated Controlnet can be updated once we start to train the model.

Denote the input feature map by $\boldsymbol{x}$, the frozen `SD Block` in Fig. 2 by $\mathcal{F}_{\mathrm{SD}}$, the extra condition by $c$, two zero convolution operators by $\mathcal{Z}^1_{\theta_1}(\cdot)$ and $\mathcal{Z}^2_{\theta_2}(\cdot)$, the trainable copy of `SD Block` by $\mathcal{G}^{\mathrm{SD}}_{\theta_s}(\cdot)$, the task instruction by $c_{\mathrm{task}}$, and the task-aware hyperNet by $\mathcal{H}_{\theta_{\mathcal{H}}}(\cdot)$. Then the output of the new task-aware modulated Controlnet can be expressed as

$$\boldsymbol{y}_c = \mathcal{F}_{\mathrm{SD}}(\boldsymbol{x}) + \mathcal{Z}^1_{\theta_1}(\mathcal{G}^{\mathrm{SD}}_{\theta_s}(\boldsymbol{x} + \mathcal{Z}^2_{\theta_2}(c) \cdot \mathcal{H}_{\theta_{\mathcal{H}}}(c_{\mathrm{task}}))) \cdot \mathcal{H}_{\theta_{\mathcal{H}}}(c_{\mathrm{task}}). \tag{1}$$

**Property of Zero Initialization.**  Similar to ControlNet [21], the weights and biases of the convolution layers are initialized as zeros. As a result, we have $\mathcal{Z}^1_{\theta_1}(\cdot) \equiv 0$ and $\boldsymbol{y}_c = \mathcal{F}_{\mathrm{SD}}(\boldsymbol{x})$, regardless of the initialization of $\mathcal{H}_{\theta_{\mathcal{H}}}(\cdot)$.

**Gradient Analysis.** We analyze the gradient of the modulated part

$$\nabla_\theta \left( Z_{\theta_1}^1(I) \cdot \mathcal{H}_{\theta_\mathcal{H}}(c_{\text{task}}) \right) = \mathcal{H}_{\theta_\mathcal{H}}(c_{\text{task}}) \cdot \nabla_\theta Z_{\theta_1}^1(I) + Z_{\theta_1}^1(I) \cdot \nabla_{\theta_\mathcal{H}} \mathcal{H}_{\theta_\mathcal{H}}(c_{\text{task}}), \qquad (2)$$

where $I$ is the input of the zero convolution layer.

When we start to train the network, the first part of the RHS of (2) follows similar analysis of ControlNet [21] since $\mathcal{H}_{\theta_\mathcal{H}}(c_{\text{task}})$ is constant when we analyze the gradient $\nabla_\theta Z_{\theta_1}^1(I)$. Since the parameters of $\mathcal{H}_{\theta_\mathcal{H}}(c_{\text{task}})$ are not initialized to zero, it is known that $\mathcal{H}_{\theta_\mathcal{H}}(c_{\text{task}}) \neq 0$. So the gradient dynamic follows the analysis of ControlNet. Therefore, we conclude that $Z_{\theta_1}^1(I) \neq 0$ after the first gradient update, and that the network can start to learn and update the following standard dynamics of stochastic gradient descent.

As for the second part of the RHS of (2), $Z_{\theta_1}^1(I) \equiv 0$ before the first gradient update, so the gradient is zero for $\theta_\mathcal{H}$. However, after the first gradient update of $\theta_1$, we know $Z_{\theta_1}^1(I) \neq 0$, and $\theta_\mathcal{H}$ can be updated with non-zero gradients.

To conclude, the new task-aware Modulated ControlNet can still be efficiently updated and learned even if the convolution layers are initialized to zero.

## C  Zero-shot-task Results and Analysis

We show more zero-shot-task results in this section, where the tasks have not been trained on. In Fig. 13, we show zero-shot deblurring results guided by the keywords. Our deblurred images can successfully recover the fine-grained details of the images without training on such data. We note that some details are still missing, *e.g.,* the details in the painting in the first row are still not clear enough. In Fig. 14, we illustrate two zero-shot image colorization results. We believe that most parts of the generated images are acceptable, though the clothes of the second woman do not look the same to the input blurred image. In Fig. 15, we observe impressive zero-shot inpainting results. In the first row, the duck that is inputted in the text has been successfully generated in the inpainted image. The second row obtains acceptable results as well, though the faces do not look perfect. The overall zero-shot quality of UniControl is remarkable.

While inpainting and outpainting might appear related, they are fundamentally distinct. Inpainting heavily leverages the contextual information from unmasked regions, necessitating a precise match. Conversely, outpainting has more freedom, with the generative model prioritizing prompts to envision new content. As shown in Fig. 8, directly using outpainting model for inpainting tasks can be challenging since the model tends to leave a sharp change over the mask boundaries. Our pretrained UniControl, thanks to intensive training across multiple tasks, has learned edge and region-to-image mappings, which assists in preserving contextual information.

Our model also demonstrates a promising capacity to generalize under scribble conditions, showing parallels to the ControlNet's ability, even though UniControl hasn't been directly trained using scribble data. Fig. 9 provides results illustrating the scribble-to-image generation.

## D  Details of User Study

In the evaluation steps, we use Amazon Mechanical Turk (Mturk) [2] to perform user study. Specifically, we ask three Mturk master workers to select the best output result for each input condition. As shown in Fig. 10, we provide instructions on guidelines to select the best generated image. The annotators are provided the condition map and the text that describes the image, and are required to select the better output between the two generated images. Considering that images can both in good or bad qualities, we provide the tie option as well. We use the majority vote to determine the result of each image, which means that an image is considered as a better image if two or more annotators vote for it. We use 294 images for the tasks of Canny, HED, Surface Normal, Depth, Segmentation, User Sketch, and Outpainting. We adopt 100 images for the task of Human Skeleton and 187 images for the task of Bounding Box. In summary, we totally obtain 7,035 voting results for all nine tasks. 2/3 of source images in testing set are collected from MSCOCO with the remaining 1/3 from Laion. And it includes a very diverse range of topics including indoor scene, outdoor scene, oil painting, portrait, pencil sketch, animation, cartoon, etc.

---

[2]https://www.mturk.com

| Input (Masked Image) | Ours-Single-Outpainting | UniControl |
| --- | --- | --- |

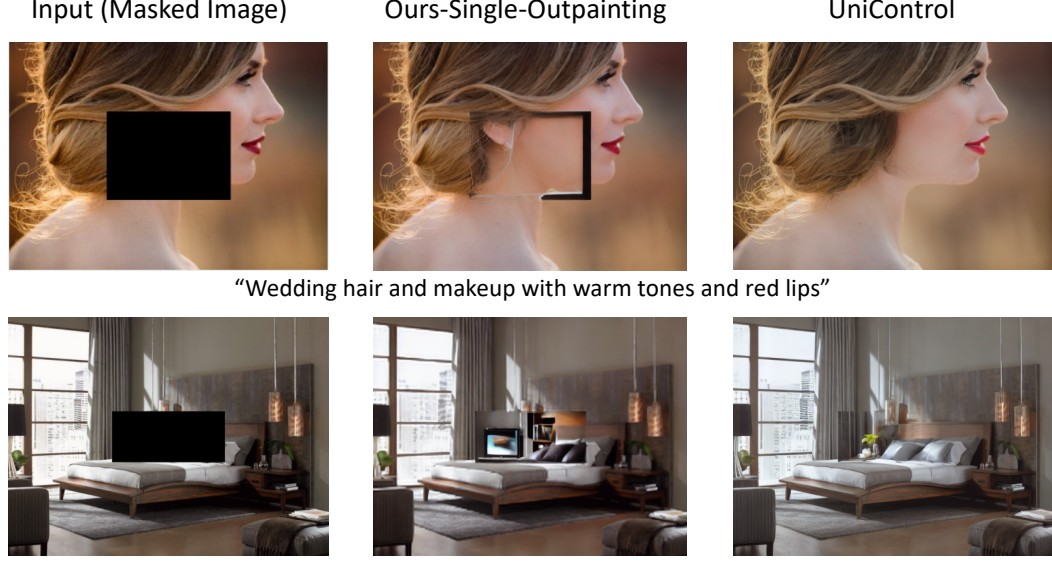

"Wedding hair and makeup with warm tones and red lips"

"Contemporary Bedroom Designs 2015 modern bedroom designs intended design"

Figure 8: Visual comparison of Ours-single-outpainting and UniControl on the inpainting task. The single outpainting model cannot well address the zero-shot inpainting task whereas UniControl demonstrates promising capacity.

| Input (Scribble) | ControlNet-Scribble Results | UniControl Results (zero-shot) |
| --- | --- | --- |

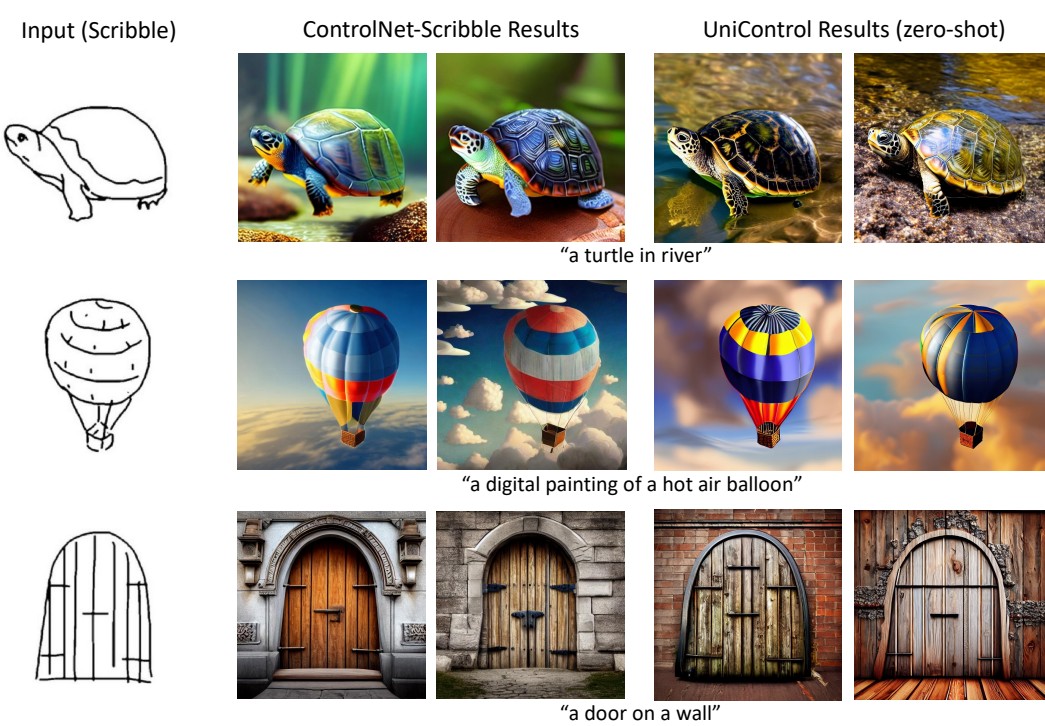

"a turtle in river"

"a digital painting of a hot air balloon"

"a door on a wall"

Figure 9: Visual comparison of ControlNet-Scribble and UniControl on the scribble data. ControlNet-Scribble is trained by the scribble data which, however, are unseen for UniControl.

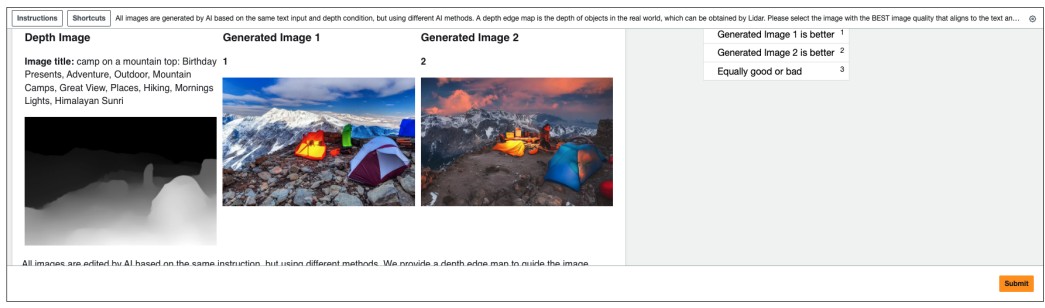

Figure 10: Mturk interface to select the better generated image.

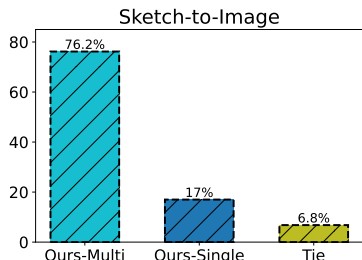

Figure 11: User study results of User Sketch to image generation.

# E Failure Cases

We illustrate some failure cases in Fig. 12. In the first row, although our generated image successfully aligns the Bounding Box condition, the generated human has a distorted body. In the second row, our generated image looks similar to the ground truth; however, the human faces are blurred. We think that the reason is that UniControl inherits the data and model bias of Stable Diffusion, where the generated human commonly have issues. In the third row, the generated image does not look realistic. We believe that the training data can be improved both quantitatively and qualitatively.

# F Additional Results

We illustrate more visualized results in this section on tasks Canny (Fig. 16), HED (Fig. 17), Depth (Fig. 18), Surface Normal (Fig. 19), Human Skeleton (Fig. 20), Bounding Box (Fig. 21), Segmentation (Fig. 22) and Outpainting (Fig. 23). These results further demonstrate the effectiveness of our proposed method. Moreover, due to the space limitation in the main paper, we report results of the last task, User Sketch. Given a sketched image, UniControl is able to achieve promising realistic images. The visualized results are in Fig. 24. The user study result can be found in Fig. 11, where it is observed that UniControl obtains significantly more votes than the single task model.

Visual Condition    Our Result    Ground Truth

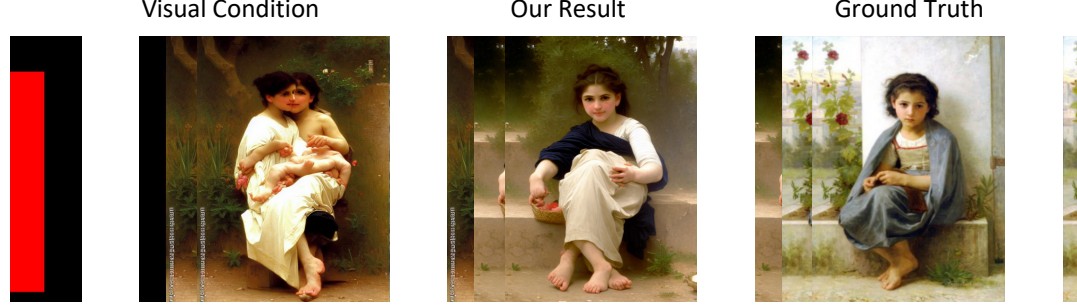

"La tricoteuse Realism William Adolphe Bouguereau Oil Paintings"

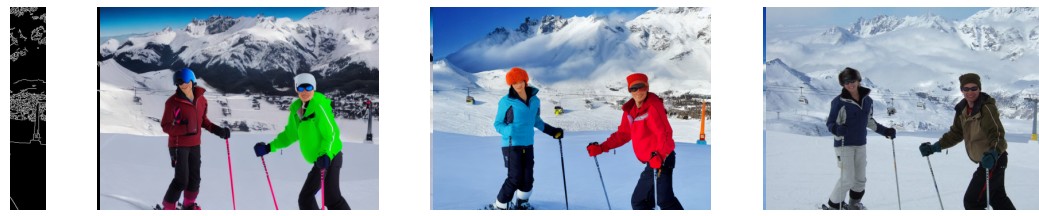

"A man and woman in ski gear standing in front of a mountain. "

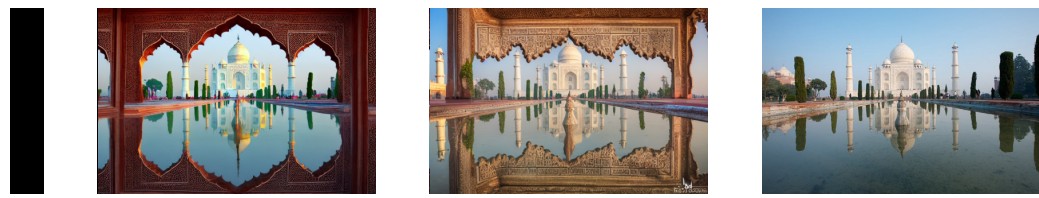

"The Taj Mahal mirrored by a water fountain's reflection. - Agra, Uttar Pradesh, India - Daily Travel Photos"

Figure 12: Failure Cases: distorted body (row one); blurred faces (row two); incorrect creation (row three).

| Blurred Image | Our Result |
| :---: | :---: |
| 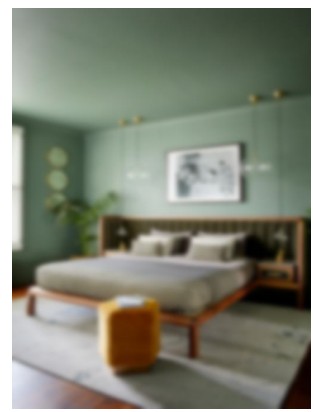 | 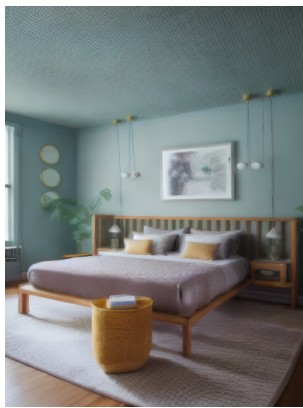 |

"Bedroom Colour Ideas 25 Paint Colours With Impact Living"

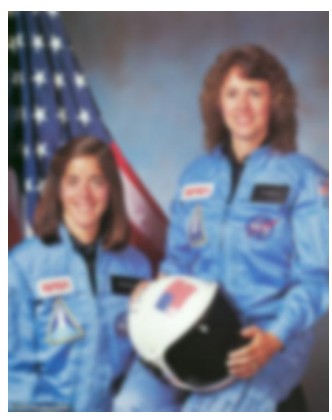 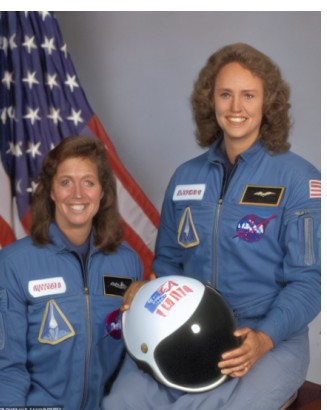

"Christa McAuliffe (right, sat with her backup crew member Barbara Morgan) was a social studies teacher who had won NASA's Teacher in Space contest and earned herself a spot on the mission"

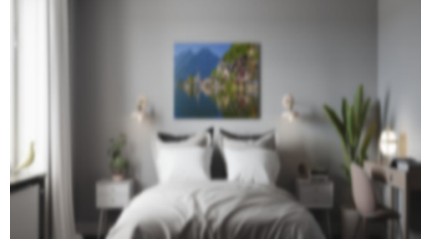 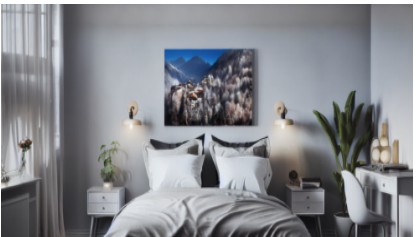

"Mountain Village in the Alps - Canvas print – Bedroom"

Figure 13: More zero-shot-task deblurring results.

|                        Gray Image                        |                        Our Result                        |
| :---: | :---: |
| 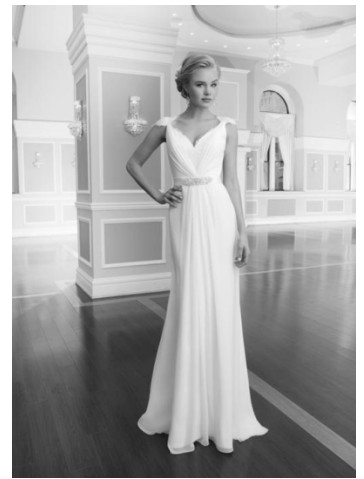 | 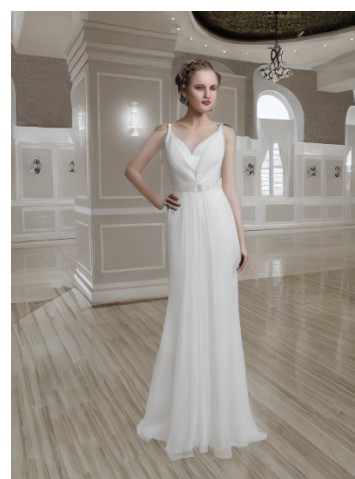 |

"Long White Casual Wedding Dress"

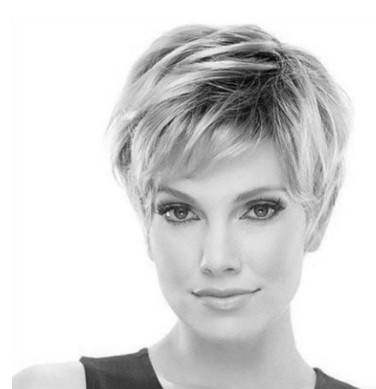 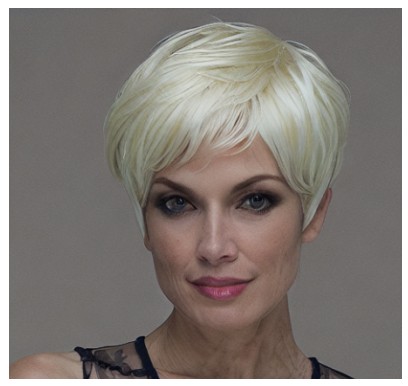

"Pixie Cropped Short Layered Synthetic Wig for Women-KAMI WIGS"

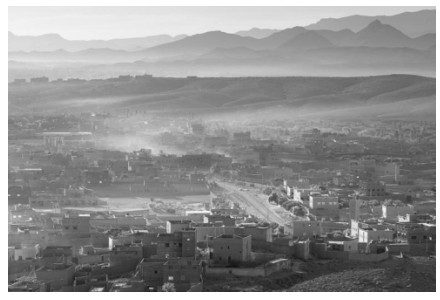 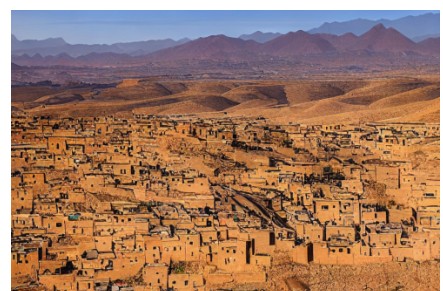

"Early morning view over the town of Tinerhir, south of the Todra Gorge, Morocco, North Africa, Africa"

Figure 14: More zero-shot-task gray-to-RGB colorization results.

Cropped Image                                        Our Result

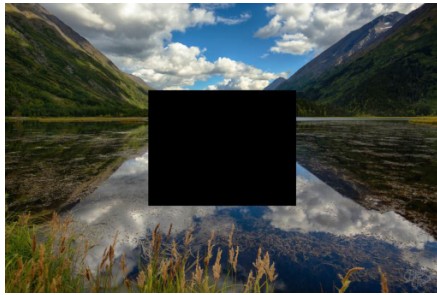    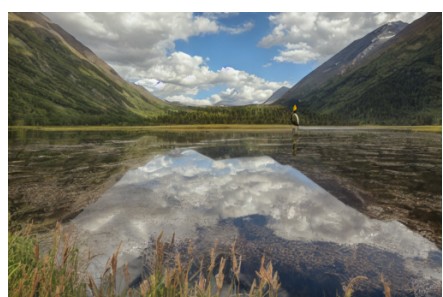

"A lone duck basks in the calm lake's mirror reflection of the Chugach mountain valley"

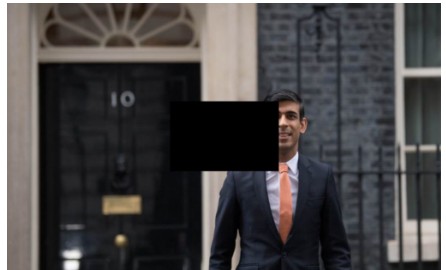    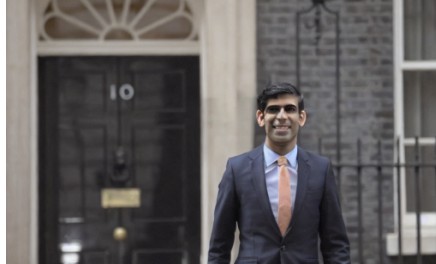

"Chancellor of the Exchequer Rishi Sunak was the most high-profile, and unexpected, appointment of the day"

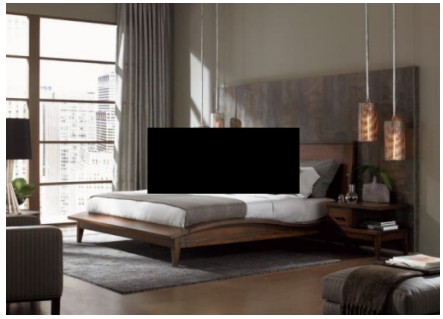    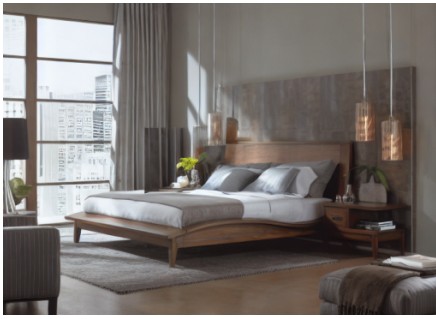

"Contemporary Bedroom Designs 2015 modern bedroom designs intended design "

Figure 15: More zero-shot-task image in-painting results. The in-painting MOE adapter weights are directly inherited from outpainting.

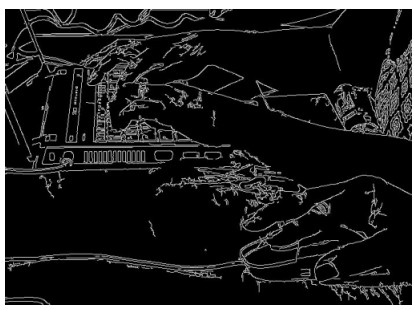
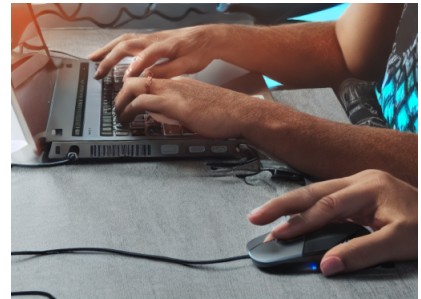

Input Image            Our Method Output

(a) "Two arms typing on a laptop and one hand on a mouse"

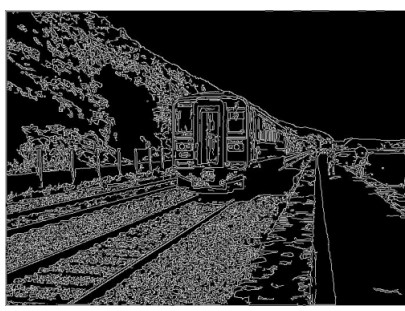
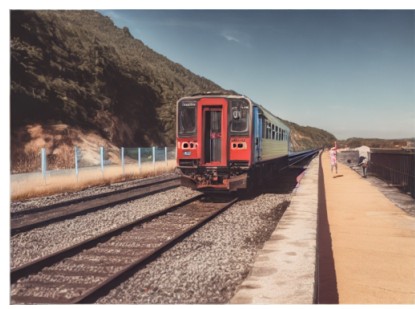

Input Image            Our Method Output

(b) "Two people walking along a side walk next to a train on the tracks."

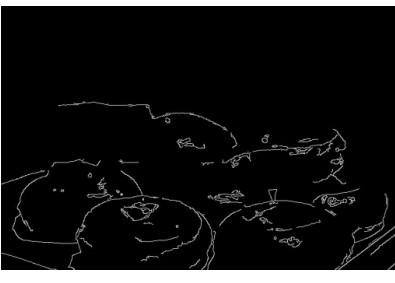
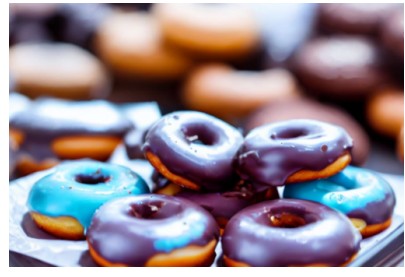

Input Image            Our Method Output

(c) "A close up of glazed donuts that are plain or with chocolate."

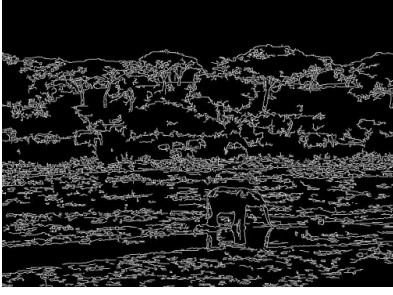
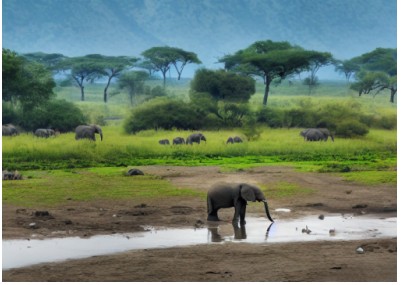

Input Image            Our Method Output

(d) "A group of elephants with water in front and trees behind."

Figure 16: Canny to Image Generation

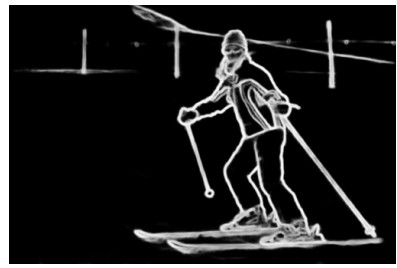

Input Image                    Our Method Output

(a) "A person on skis makes her way through the snow"

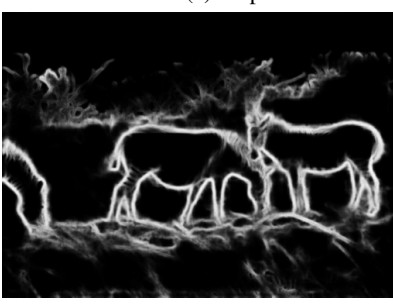 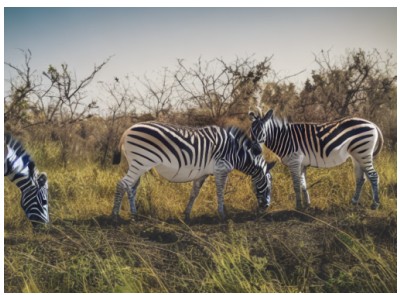

Input Image                    Our Method Output

(b) "Three zebras grazing in a grassy area near shrubs"

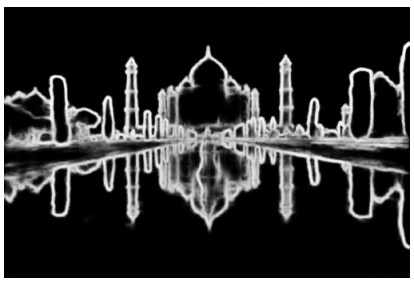 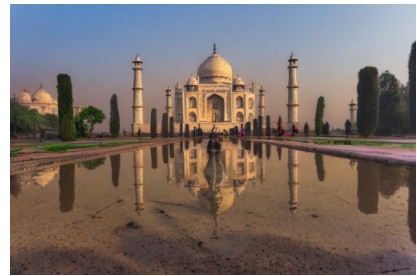

Input Image                    Our Method Output

(c) "The Taj Mahal mirrored by a water fountain's reflection. - Agra, Uttar Pradesh, India - Daily Travel Photos"

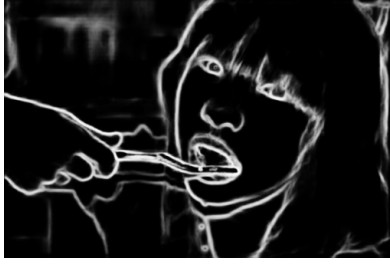 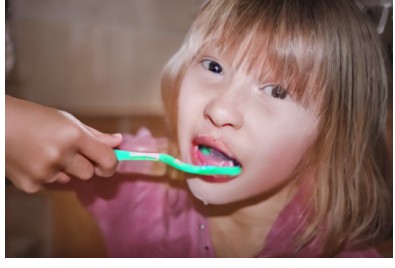

Input Image                    Our Method Output

(d) "A young girl who is brushing her teeth with a toothbrush."

Figure 17: HED to Image Generation

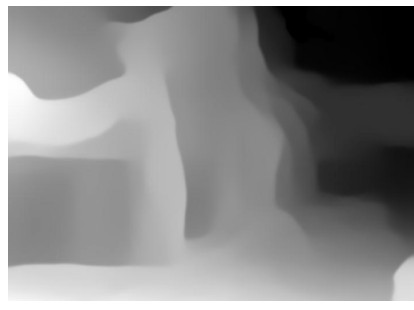
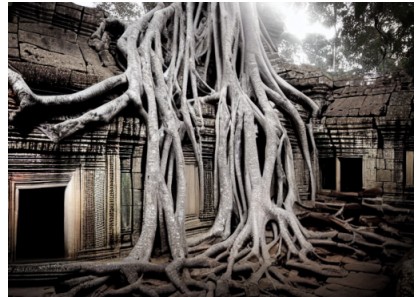

Input Image            Our Method Output

(a) "The Ta Prohm Temple Located at Angkor in Cambodia by Kyle Hammons"

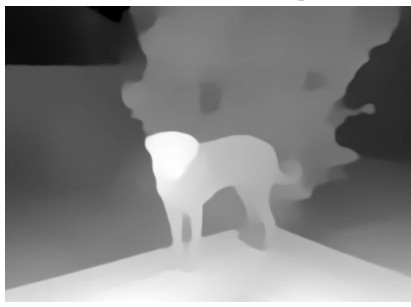
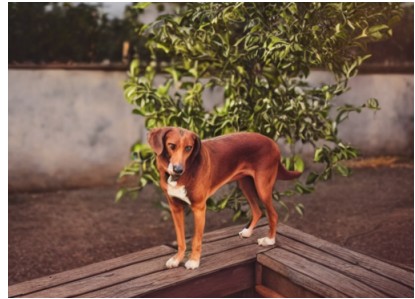

Input Image            Our Method Output

(b) "A brown dog standing on a wooden bench near a lemon tree."

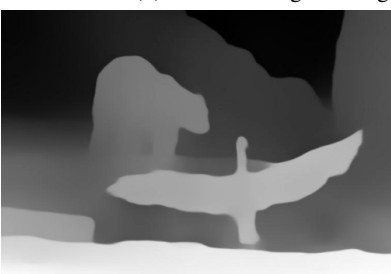
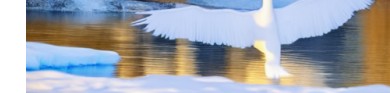

Input Image            Our Method Output

(c) "A Polar Bear walks toward water, while a large bird lands on the opposite bank."

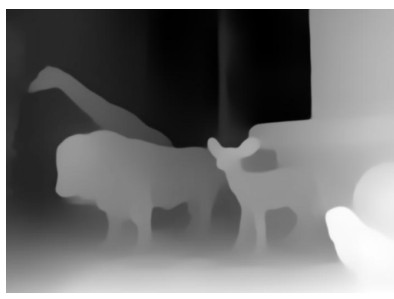
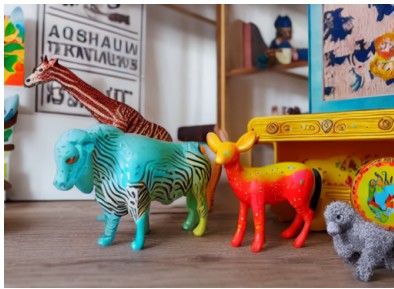

Input Image            Our Method Output

(d) "A display of vintage animal toys on the floor."

Figure 18: Depth to Image Generation

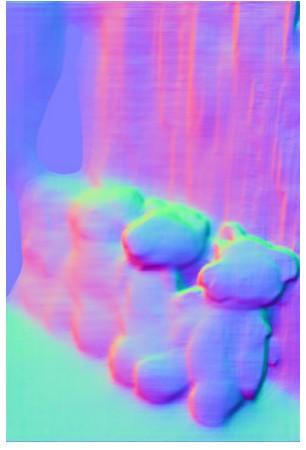 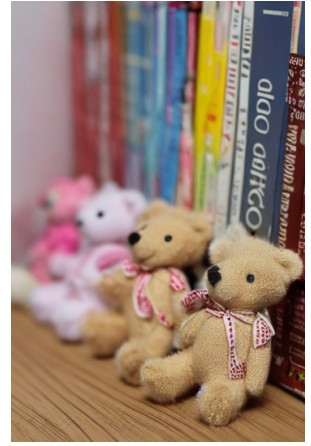

Input Image                              Our Method Output

(a) "A line of small teddy bears are in front of several DVD cases."

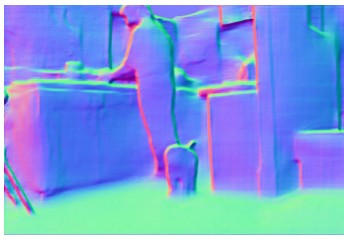 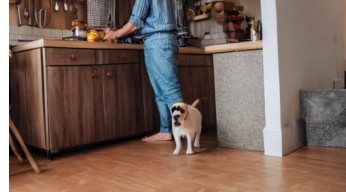

Input Image                              Our Method Output

(b) "A man in the kitchen standing with his dog"

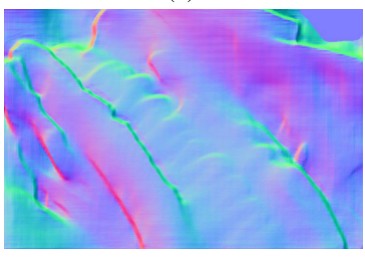 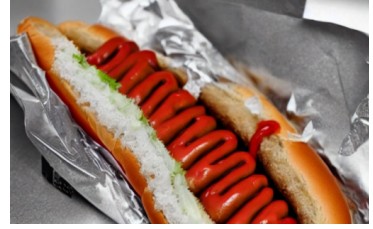

Input Image                              Our Method Output

(c) "A hot dog sitting on top of a bun in a wrapper"

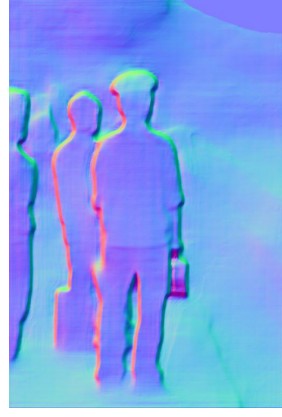 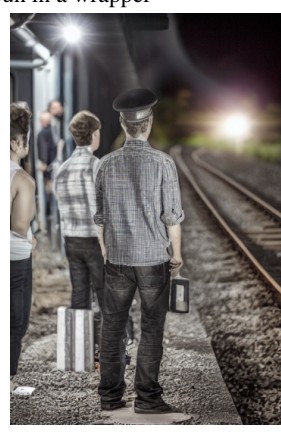

Input Image                              Our Method Output

(d) "Several people waiting on the side of train tracks as a train with it's lights on comes down the track"

Figure 19: Surface Normal to Image Generation

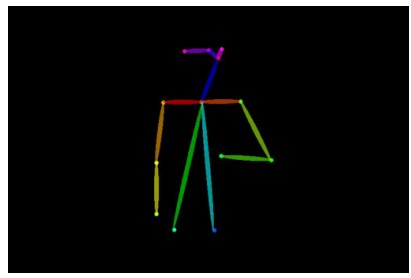

Input Image                                    Our Method Output

(a) "Photo of handsome man in black leather jacket"

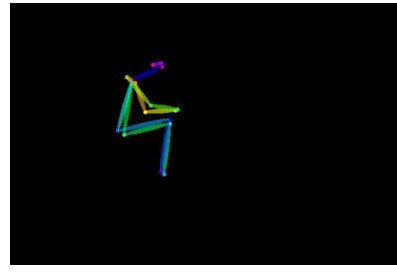 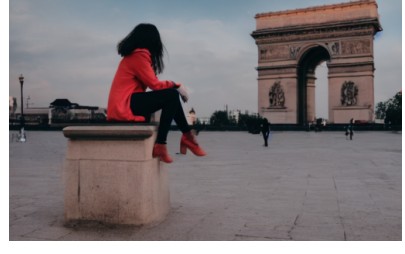

Input Image                                    Our Method Output

(b) "A woman is sitting near a prominent landmark"

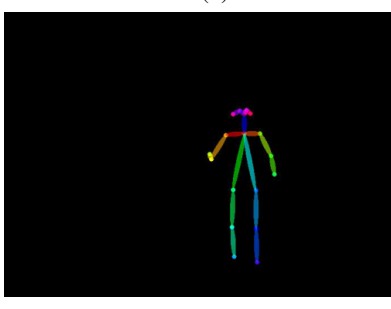 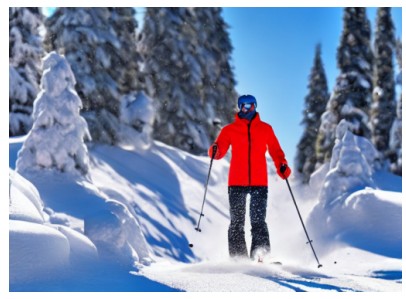

Input Image                                    Our Method Output

(c) "A man that has ski's and is standing in the snow."

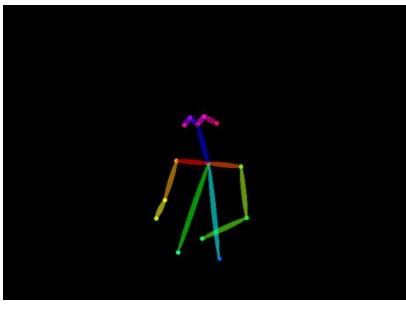 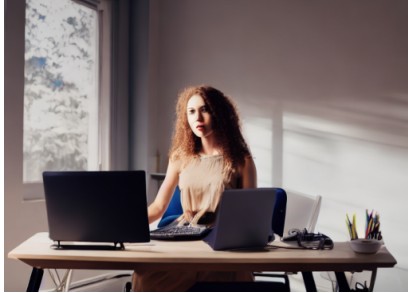

Input Image                                    Our Method Output

(d) "A woman is sitting in front of a desk"

Figure 20: Human Pose Skeleton to Image Generation

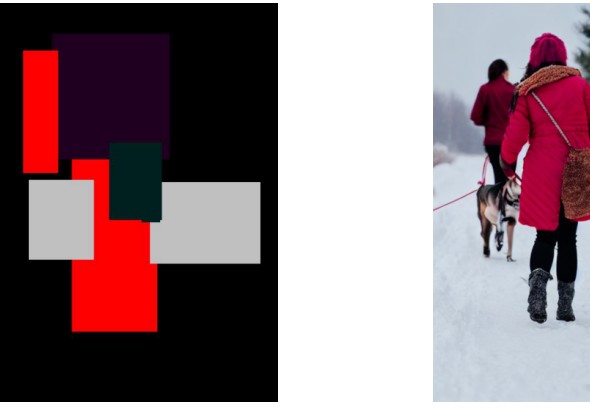

Input Image            Our Method Output

(a) "A woman is walking two dogs in the snow"

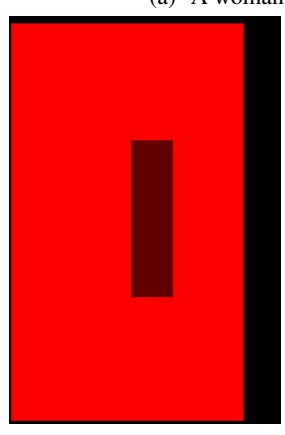 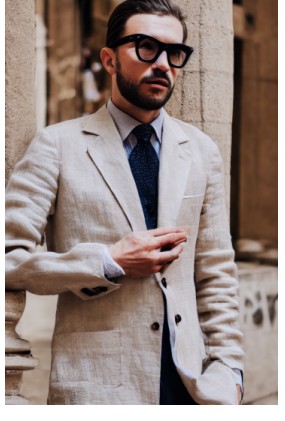

Input Image            Our Method Output

(b) "Simone Righi frasi glasses linen suit menswear streetstyle icon fashion florence."

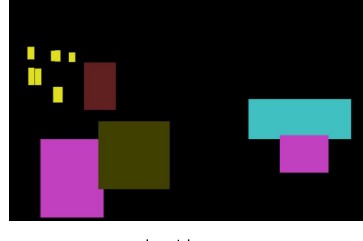 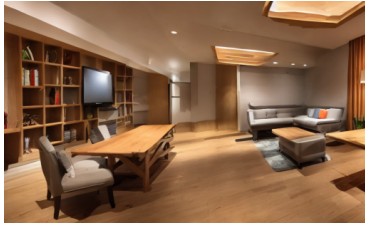

Input Image            Our Method Output

(c) "The large room has a wooden table with chairs and a couch."

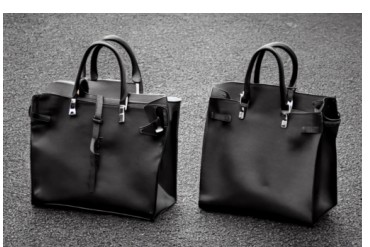

Input Image            Our Method Output

(d) "Two black bags placed standing on the ground"

Figure 21: Bounding Box (by YOLO-V4-MSCOCO) to Image Generation

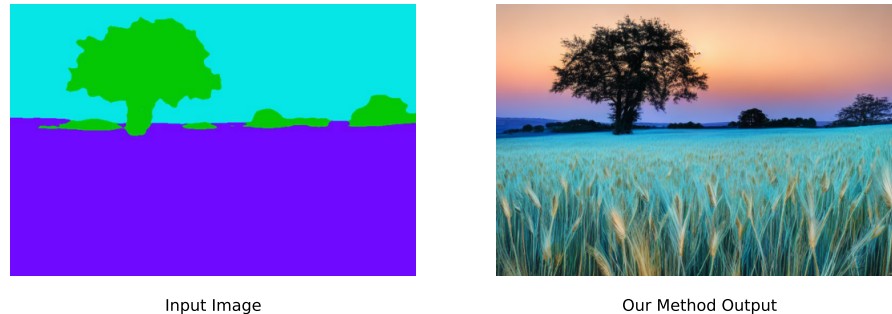

Input Image                           Our Method Output

(a) "A bench at the beach next to the sea"

Input Image                           Our Method Output

(b) "Water traffic along the Thames by Big Ben"

Input Image                           Our Method Output

(c) "A well-lit and well-decorated living room shows a glimpse of a glass front door through the corridor."

Input Image                           Our Method Output

(d) "Blue Hour Barley"

Figure 22: Segmentation Map (by Uniformer-ADE20K) to Image Generation

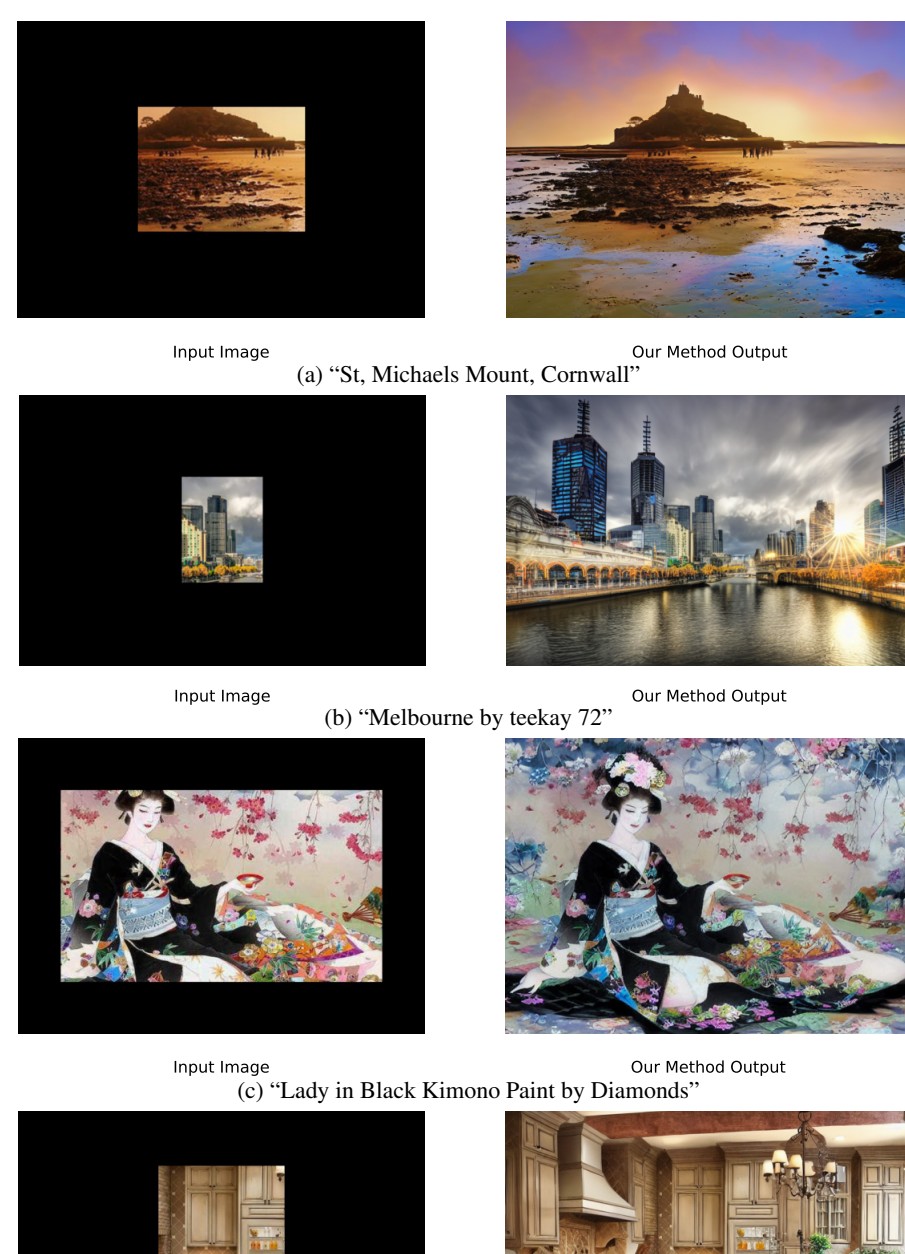

Input Image                         Our Method Output
(a) "St, Michaels Mount, Cornwall"

Input Image                         Our Method Output
(b) "Melbourne by teekay 72"

Input Image                         Our Method Output
(c) "Lady in Black Kimono Paint by Diamonds"

Input Image                         Our Method Output
(d) "Beautiful kitchen grand scale living pinterest for Kitchen cabinets lowes with old world metal wall art"

Figure 23: Image Outpainting

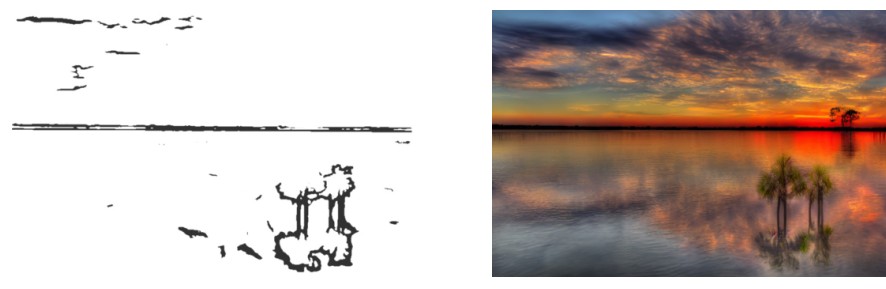

Input Image                                      Our Method Output

(a) "A Limited Edition, Fine Art photograph of a beautiful sunrise at Lake Jackson in Sebring, Florida. Available as a Fine Art print"

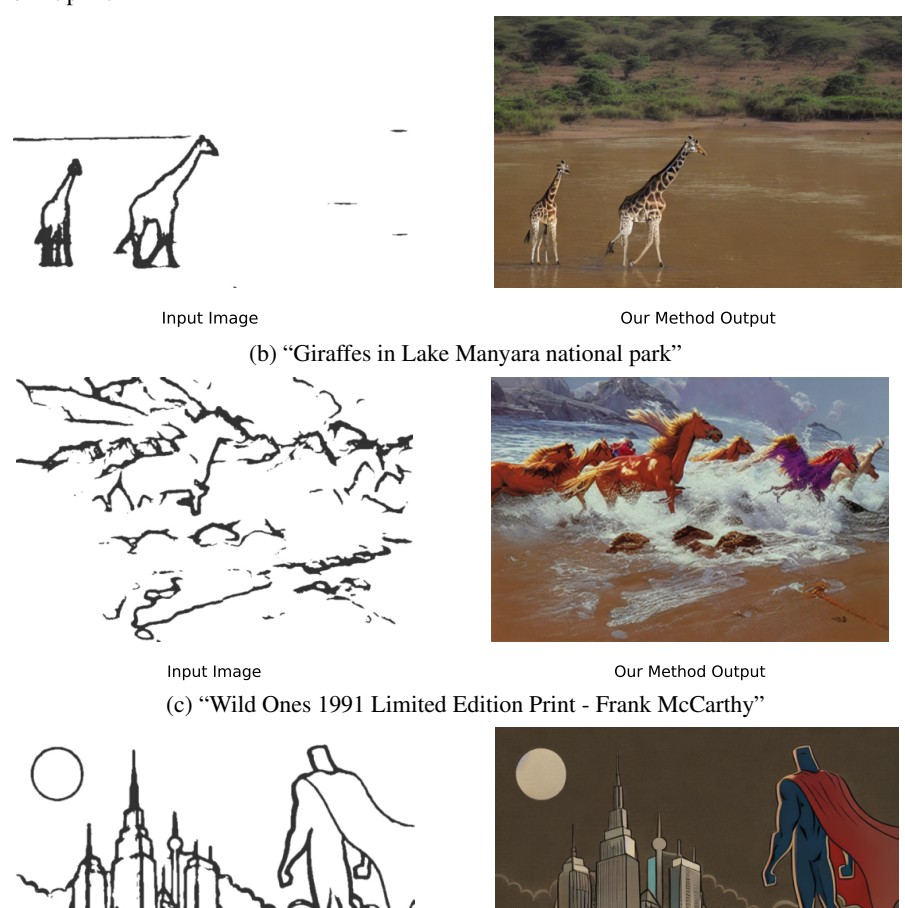

Input Image                                      Our Method Output

(b) "Giraffes in Lake Manyara national park"

Input Image                                      Our Method Output

(c) "Wild Ones 1991 Limited Edition Print - Frank McCarthy"

Input Image                                      Our Method Output

(d) "Superhero watching over city. No transparency used. Basic (linear) gradients. A4 proportions."

Figure 24: User Sketch to Image Generation

