# OpenReview forum: "UniControl: A Unified Diffusion Model for Controllable Visual Generation In the Wild"
_NeurIPS.cc/2023/Conference — NeurIPS 2023 poster_

### Official Review · Reviewer_HFTZ · 2023-07-04

**Soundness:** 2 fair
**Presentation:** 2 fair
**Contribution:** 2 fair
**Rating:** 5
**Confidence:** 4

**Summary:**

The paper proposes a unified diffusion framework, UnitControl for a more fine-grained controllable generation based on input visual conditions, utilizing two modules mixture-of-experts adaptor that extract features from different visual conditions, and a task-aware HyperNet that extracts language-based task embedding to control visual generation.

**Strengths:**

1. The paper conducts a relatively comprehensive evaluation of the proposed method, including human evaluations.
2. The framework can be useful as it unified different modalities for visual controllable generation using diffusion models.
3. The authors promise to open-source 20M triple training datasets that can benefit the whole community.

**Weaknesses:**

1. **The writing is not so well written and missing details that can hinder understanding of readers**. For example, Figure 2 should also mention parameters of encoder and decoder since I initially thought the introduced parameters are only from MOE adaptors and Hypernet. The authors should also highlight the differences and **similarities** (e.g., encoder and decoder are copied) between their work and ControlNet. In Section 3.3, how is hybrid task generalization done? Since your model only conditions on one visual modality when computing the score in one forward pass, do you compute scores individually for each condition for generation? If so, what sampling technique did you use? One sampling technique I can think of is the conjunction operator from Composable Diffusion [1] or do you use some other existing techniques? For zero-shot task generalization, I also find it confusing and don't quite understand how it is done. Mentioning it in the supplementary is not sufficient since it is meant to provide details that don't affect your understanding overall.

2. **The claims are quite shaky.** The authors claim that the model need to tackle the misalignment of low-level features from different tasks. However, based on Figure 4, I don't see too much of misalignment from baselines, i.e., ControlNet. Besides the visual quality, I don't see much of advantages over ControlNet. Even sometimes single-task ControlNet is better in some aspects. For example, for the object bounding box to image, control net seems to understand there should be two separate benches instead of one long bench generated by the proposed model. In addition, since one unified model is trained for longer time, model could be more robust, directly improving overall visual quality.

3. **The zero-shot generalization ability could be misleading.** Suppose in training, there doesn't exist any instructions that perform colorization. I don't expect the language itself can bridge the gap between visual signals and language to perform such unseen tasks. For example, in Figure 5, such zero-shot results doesn’t necessarily reflect the true reality that the model understands given instructions. One way to check this is to reverse the whole process - for example, can you do image decolorization using your prompt or blur the image instead. Since training images are mostly high resolution, and colored images, the model could automatically utilize learned prior to make images higher quality, which doesn’t necessarily mean they are doing what they are told to do. If so, then the model just does general reconstruction when prompted something unknown.

4. **Missing quantitative evaluations.** As stated, one contribution is to show that the visual quality of such controllable generation can be improved, then it is needed to include quantitative comparisons for image fidelity using metrics such as FID and KID, though I don't think they are good metrics but it provides enough insights.

5. **Lacking relevant baselines.** There are also other existing adaptor methods for such generation, for example, T2I-adaptor [2] and GLIGEN [3] seem to be highly relevant, and both of them are not used as a comparison in the paper.

5. **Related work.** If one of the contributions is to combine different modalities for compositional generation, then it would be important to add existing works related to compositional generation.

[1] Liu et al., Compositional Visual Generation with Composable Diffusion Models (ECCV 2022) \
[2] Mou et al., T2I-Adapter: Learning Adapters to Dig out More Controllable Ability for Text-to-Image Diffusion Models. \
[3] Li et al., GLIGEN: Open-Set Grounded Text-to-Image Generation (CVPR 2023)

**Questions:**

1. How are both hybrid tasks generalization and zero-shot new task generalization done? For example, a detailed procedure (i.e., sampling technique) will be useful for better understanding.
2. There are 6 tasks overlapped with ControlNets, each of which is trained for 100K iterations. However, there are 8 tasks in your multi-task model, but it is trained for 900K, which in fact is not a fair comparison. If I misunderstood, feel free to point that out.
3. The method mainly uses modalities that are spatial, so is it capable of using global context, e.g., color, texture, to guide image generation?

**Limitations:**

The authors have included limitations and broader impact such as training data bias.

---

> ### Author Rebuttal · Authors · 2023-08-10
>
> We sincerely appreciate your suggestions and questions of our paper. Your concerns are addressed as follows.
>
> **Q1:  UniControl vs ControlNet**
>
> Components and #params of whole UniControl Model and Multi-ControlNet
> |  | Stable Diffusion | ControlNet  | MoE-style Adapters | TaskHyperNet | Total|
> |--|--|--|--|--|--|
> | UniControl| 1065.7M  | 361M | 0.06M | 12.7M | 1.44B|
> | Multi-ControlNet| 1065.7M  | 361M * 9  |- | - | 4.315B|
>
>
> Compared with the original ControlNet Model (Stable Diffusion + ControlNet)), the increase in UniControl's size is  0.09%, amounting to an additional 12.76M parameters. Notably, UniControl's versatility spans nine distinct tasks. In contrast, Multi-ControlNet (assembly of multiple ControlNets) needs a single Stable Diffusion and nine separate ControlNets to achieve comparable results. UniControl (1.44B #params)  has greatly reduced the complexity compared with the Multi-ControlNet (4.315B #params) to achieve the same goal with better generation quality. The Stable Diffusion Model (including U-Net, CLIPText, VQVAE) is directly copied from its official checkpoint that is the same as the ControlNet's Stable Diffusion part.
>
>
>
> **Q2:  Details of zero-shot hybrid and new task inference**
>
> Thank you for your suggestions. The work presented in [1] indeed offers a novel perspective on compositional generation via multiple conditions. However, UniControl adopts a distinct approach, detailed as follows:
>
> *Hybrid Task:* As illustrated in the left subfigure of Fig. 3, the features from two condition inputs are integrated through an addition operation. This fused feature set is consolidated into a single tensor, which is subsequently fed into the ensuing ControlNet modules that are interleaved with modulated zero-convs. The entire inference procedure follows the standard DDIM protocol the same as ControlNet and single task UniControl.
>
> *New Task:* Similar to the hybrid task, we feed the novel condition into a weighted ensemble of MoE-style adapters. An example configuration might be “depth: 0.6, seg: 0.3, canny: 0.1” for the colorization task. This process yields one feature tensor that is fed to subsequent modules. Again, the inference follows ControlNet's standard methodology, employing the regular DDIM.
>
> We appreciate the reference to [1] and plan to incorporate it into our forthcoming manuscript.
>
>
> **Q3: More quantitative evaluations and baseline methods**
>
> Thank you for your suggestion. We've conducted quantitative analysis to include classic baselines such as GLIGEN [1], T2I-adapter [2], and ControlNet [3]. Our experimental setup remains consistent with Sec. 4.3 of the main manuscript, utilizing DDIM as the sampler with a guidance score of 9. With a collection of over 2,000 test samples sourced from Laion and COCO, we've assessed a wide range of tasks covering edges (Canny, HED), regions (Seg), skeletons (Pose), and geometric maps (Depth, Normal).
>
>
> FID Scores
> |  | GLIGEN | T2I-adapter  | ControlNet | UniControl |
> |--|--|--|--|--|
> | Canny | 24.9 | 23.6 | **22.7** | 22.9 |
> | HED | 27.8 | - | 25.1 | **23.6**  |
> | Depth | 25.8 | 25.4 | 25.5 | **21.3** |
> | Normal | 27.7 | - | 28.4 | **23.4** |
> | Seg | - | 27.1 | 26.7 | **25.5** |
> | Pose | - | 28.9 | 28.8 | **27.4** |
>
> [1] GLIGEN: Open-Set Grounded Text-to-Image Generation. CVPR 23.
>
> [2] T2I-Adapter: Learning Adapters to Dig out More Controllable Ability for Text-to-Image Diffusion Models. arXiv 2302.08453.
>
> [3] Adding Conditional Control to Text-to-Image Diffusion Models. arXiv 2302.05543.
>
>
>
> **Q4: Claims are shaky.**
>
> ControlNet trains separate models for different tasks, thus there is no parameter sharing for each task, and no misalignment for ControlNet. MoE-adapter is specifically designed to solve the misalignment issue for UniControl. Notably, UniControl's versatility spans nine distinct tasks. In contrast, Multi-ControlNet (assembly of multiple ControlNets) needs a single Stable Diffusion and nine separate ControlNets to achieve comparable results. UniControl (1.44B #params)  has greatly reduced the complexity compared with the Multi-ControlNet (4.315B #params) to achieve the same goal with better generation quality.
>
>
> For the bench example you mentioned (Fig. 4), the prompt is “A women sitting on a bench near a statue, checking her phone”. UniControl generates a real woman checking her phone, while ControlNet generates two statues, not checking her phone.
>
>
>
> **Q5: Zero-shot generalization ability is misleading**
>
> Directly using outpainting model for inpainting tasks can be challenging since the model tends to leave a sharp change over the mask boundaries as shown in pdf file. Currently we think the generalization ability mainly comes from the combination of different visual conditions by assigning different weights of the visual input. If we want to have more stronger generalization ability on zero-shot tasks, we agree we need to specifically train it.
>
> **Q6: More related works [1,2,3]**
>
> Thanks for pointing out the highly related works, we included [2,3] for comparison and will cite and discuss [1] in our upcoming version.
>
>
>
> **Q7: Fair comparison with ControlNet in training cost**
>
> Actually, the iterations are not fair to compare since the different configurations of model training. Instead, we compare the training cost on the GPU Hours. UniControl is trained by ~5000 GPU Hours by A100-40G. This is comparable to the overall training cost of different ControlNets with ~2500 GPU Hours on A100-80G.
>
> |  | Canny | HED | Pose | Seg| Depth| Normal|Sketch | Total|
> |--|--|--|--|--|--|--|--|--|
> | Hours |600 | 300 | 300 | 400 | 500 |  200| 150 | 2450 |
> | GPU |A100-80G  | A100-80G | A100-80G | 3090TI | A100-80G |A100-80G  | A100-80G |   |
>
>
>
>
> **Q8: Inclusion of global context such as color and texture as condition**
>
> We plan to include new conditions to UniControl similar as the T2I-Adaper-Color. Moreover, we can integrate the LoRA or Dreambooth to UniControl to control the style or texture of generative results.

---

> > ### Comment · Reviewer_HFTZ · 2023-08-18
> >
> > I greatly appreciate authors rebuttal and it does address most of my concerns.
> >
> > Please incorporate what the authors promised in the next version of the paper.
> >
> > I have raised my rating to borderline accept.

---

> > > ### Author Response · Authors · 2023-08-18
> > > **After Response**
> > >
> > > We sincerely thank the reviewer HFTZ for your detailed feedback, and are happy to hear that our rebuttal addressed your concerns! As suggested, we will certainly make revisions to the manuscript to add these quantitative comparisons and clarifications of related research works.
> > >
> > > Best regards,
> > > Authors of 6000

---

### Official Review · Reviewer_y5r6 · 2023-07-05

**Soundness:** 3 good
**Presentation:** 3 good
**Contribution:** 2 fair
**Rating:** 5
**Confidence:** 3

**Summary:**

UniControl is a diffusion-based image generation model that can condition on natural language input as well as multiple types of visual inputs (e.g. edge map, depth map). The framework is built upon components of Stable Diffusion Models, an MOE adapter and a ControlNet modulated by a task-aware HyperNet. The model demonstrates abilities of conditioning on one or multiple visual inputs at a time, as well as visual inputs that it has not seen during training.

**Strengths:**

- UniControl extends ControlNet to work with multiple tasks and shows that the tasks help each other so as to improve performance on single task metrics as well.
- the MOE adapter set up easily allows the model to condition on multiple visual inputs.

**Weaknesses:**

- The model has a certain complexity, as it involves multiple modules such as the MOE adapter, the ControlNet and a HyperNet.
- It needs to maintain two sets of SDM parameters. (This is the same as ControlNet, though.)
- MOE adapter means more parameters to be added with each new task added.
- Task instruction is needed for both training and inference, and is handled by a separate module than the language prompt, even though both are text prompts. This can be a downside if the task instruction is unknown or not well defined.
- It is not clear what $c_{task}$ is used for the task-aware hypernet during hybrid-task or zero-shot new task inference.

Ablation study would be helpful to show the importance/usefulness of the proposed components:
- model performance without the MOE adapter.
- model performance without conditioning on $c_{task}$, or merge $c_{task}$ into $c_{text}$ (in a thoughtful way). This will show whether the task-aware hypernet is needed.

**Questions:**

- HyperNet should be cited?

**Limitations:**

The authors have discussed the limitations and broader impact of this work.

---

> ### Author Rebuttal · Authors · 2023-08-10
>
> We are grateful for your suggestions to enrich our paper. Your concerns and questions are addressed as follows.
>
> **Q1:  Complexity of UniControl**
>
> Components and #params of whole UniControl Model and Multi-ControlNet
> |  | Stable Diffusion | ControlNet  | MoE-style Adapters | TaskHyperNet | Total|
> |--|--|--|--|--|--|
> | UniControl| 1065.7M  | 361M | 0.06M | 12.7M | 1.44B|
> | Multi-ControlNet| 1065.7M  | 361M * 9  |- | - | 4.315B|
>
> Compared with the original ControlNet Model (incorporating Stable Diffusion + ControlNet), the complexity of UniControl has seen a marginal increase of 0.09%, amounting to an addition of 12.76M parameters. Despite this modest augmentation, UniControl's capability extends to nine distinct tasks. In contrast, the Multi-ControlNet (an ensemble of several ControlNets) demands a separate Stable Diffusion plus nine individual ControlNets to accomplish the same tasks. This underscores the efficiency of UniControl, which at 1.44B parameters, significantly trims the complexity when compared to the 4.315B parameters of Multi-ControlNet, all the while achieving a comparable objective.
>
>
> **Q2:  MoE adapter means more parameters to be added with each new task added.**
>
> MoE-style adapters are exceptionally lightweight. It has ~0.06M #params whose size and associated computation cost are marginal compared with the entire model.
>
>
> **Q3:  Downside of task instruction**
>
> As explained in Sec. A.2 of the Appendix, we utilize a predefined set of task instructions that can be systematically matched to each respective task, ensuring a consistent and transparent process. When it comes to unknown tasks, our model demonstrates robustness towards a spectrum of new instructions. This is largely attributed to the clustering of their embeddings, due to their semantic similarities.
>
> TaskHyperNet is efficient in size, comprising just 12.7M parameters. And there are some techniques for acceleration. One such technique involves offline collection of task embeddings. By doing so, we can bypass the need for instruction-to-embedding inference and instead directly retrieve the pre-calculated task embeddings, expediting the process.
>
>
>
> **Q4:  How does Task-aware HyperNet use in the zero-shot tasks?**
>
> In the context of zero-shot tasks, we defined the task instruction in a straightforward way. Examples include designations like "image inpainting" or "segmentation map and human skeleton to image." These instructions are subsequently processed by CLIPText and the Task-aware HyperNet to derive the requisite task embeddings.
>
>
> **Q5:  Missing Ablation Study**
>
> Thank you for pointing this out. In response, we've conducted an ablation study, specifically focusing on the MoE-Style Adapter and TaskHyperNet. The table contains the FID scores. It is noticeable that the full-version UniControl (MoE-Style Adapter + TaskHyperNet) constantly outperforms the ablations.
>
>
> | MoE-Adapter | TaskHyperNet | Canny | HED | Depth | Normal  | Seg  |  Pose | Avg|
> |--|--|--|--|--|--|--|--|--|
> | x | x | 27.2 | 29.0 | 27.6 | 28.8 |29.1  | 30.2 | 28.7 |
> | ✓ | x | 24.5 | 26.1 | 23.7 | 24.8 | 26.9 | 28.3 | 25.7 |
> | ✓ | ✓ | **22.9**  | **23.6** | **21.3** | **23.4** | **25.5** | **27.4** | **24.0** |
>
> **Q6:  Cite HyperNet**
>
> Thank you for your suggestion. It is our mistake. We have updated these two references [1,2] in our latest manuscript.
>
> [1] HyperNetworks. ICLR 17.
>
> [2] Continual Learning with Hypernetworks. ICLR 20.

---

> > ### Comment · Reviewer_y5r6 · 2023-08-20
> > **Response**
> >
> > Thank you for the rebuttal. Regarding Q1, what I meant by "complexity" is not equivalent to the number of parameters. A method can be complex yet computationally cheap (e.g. it can contain a lot of lightweight components) or simple but expensive (e.g. a large-scale LLM). I still do think the method is on the complex side. With the added ablation study I am willing to raise my score to borderline accept.

---

> > > ### Author Response · Authors · 2023-08-21
> > > **After Feedback**
> > >
> > > Dear Reviewer y5r6,
> > >
> > > We sincerely appreciate your constructive comments and positive recognition of the paper. The ablation study will be added into the new version manuscript. We agreed with the point of complexity in methodology since unified controllable generation is a bright new and challenging task. Compared with the directive baseline, Multi-ControlNet, we believe the UniControl has a lower complexity in both inference and training. We will continue to explore more simple and effective solutions in this area.
> > >
> > > Best regards,
> > > Authors of 6000

---

### Official Review · Reviewer_Haxt · 2023-07-06

**Soundness:** 3 good
**Presentation:** 3 good
**Contribution:** 2 fair
**Rating:** 6
**Confidence:** 4

**Summary:**

This paper introduces UniControl, a new generative foundation model that consolidates a wide array of controllable condition-to-image (C2I) tasks within a singular framework. UniControl enables pixel-level-precise image generation, where visual conditions primarily influence the generated structures and language prompts guide the style and context. For this purpose, the authors augment pretrained text-to-image
diffusion models (ControlNet) and introduce a task-aware HyperNet to modulate the diffusion models, enabling the adaptation to different C2I tasks simultaneously. UniControl was trained on nine unique C2I task and demonstrated excellent zero-shot generation abilities with unseen visual conditions.

**Strengths:**

1. The paper is clearly written and easy to follow.
2. The related work section covers the most relevant papers in the field.
3. The approach produces excellent image generation results.
4. The experimental evaluation is convincing.

**Weaknesses:**

1. It relies too much on existing image generation methods (ControlNet).
2. A more scientific contribution would have been expected.

**Questions:**

Here are my concerns:
1. The paper claim in the abstract that: "...UniControl often surpasses the performance of single-task-controlled methods of comparable model sizes". Maybe I missed something, but could the authors elaborate more on this statement? What are the other 'single-task-controlled methods' they refer to? I only found comparison with ControlNet.
2. Could you adapt your approach to work with other Image Generation software (besides Stable Diffusion)?

**Limitations:**

Limitations are addressed in the paper in a dedicated section.

---

> ### Author Rebuttal · Authors · 2023-08-10
>
> Thank you for your valuable time to review this paper. Your concerns are addressed below.
>
>
> **Q1: More comparison with the single-task-controlled methods**
>
> Thank you for your valuable suggestion. In response, we've expanded our quantitative analysis to include classic single-task-controlled methods such as GLIGEN [1], T2I-adapter [2], and ControlNet [3]. Our experimental setup remains consistent with Sec. 4.3 of the main manuscript, utilizing DDIM as the sampler with a guidance score of 9. With a collection of over 2,000 test samples sourced from Laion and COCO, we've assessed a wide range of tasks covering edges (Canny, HED), regions (Seg), skeletons (Pose), and geometric maps (Depth, Normal).
>
> The following FID table demonstrates that our UniControl consistently surpasses the baseline methods across the majority of tasks. Notably, UniControl achieves this while maintaining a more compact and efficient architecture than its counterparts.
>
>
> FID Scores
> |  | GLIGEN | T2I-adapter  | ControlNet | UniControl |
> |--|--|--|--|--|
> | Canny | 24.9 | 23.6 | **22.7** | 22.9 |
> | HED | 27.8 | - | 25.1 | **23.6**  |
> | Depth | 25.8 | 25.4 | 25.5 | **21.3** |
> | Normal | 27.7 | - | 28.4 | **23.4** |
> | Seg | - | 27.1 | 26.7 | **25.5** |
> | Pose | - | 28.9 | 28.8 | **27.4** |
>
> [1] GLIGEN: Open-Set Grounded Text-to-Image Generation. CVPR 23.
> [2] T2I-Adapter: Learning Adapters to Dig out More Controllable Ability for Text-to-Image Diffusion Models. arXiv 2302.08453.
> [3] Adding Conditional Control to Text-to-Image Diffusion Models. arXiv 2302.05543.
>
>
> **Q2: Adapt UniControl method to work with other Image Generation software (besides Stable Diffusion)**
>
> Yes. UniControl is versatile and can be adapted to various diffusion-based models, including the likes of Deep-floyd [1]. To facilitate this integration, we project the embeddings from UniControl onto the new model's backbone using cross-attention layers or linear mapping. However, it's essential to note that this integration necessitates re-training UniControl to ensure seamless alignment with the new backbone.
>
>
> [1] https://github.com/deep-floyd/IF

---

> > ### Comment · Reviewer_Haxt · 2023-08-20
> > **Response to Rebuttal**
> >
> > I am satisfied with the authors' rebuttal, therefore I maintain my initial rating.

---

> > > ### Author Response · Authors · 2023-08-21
> > > **After Rebuttal**
> > >
> > > Dear Reviewer Haxt,
> > >
> > > Thank you for your constructive comments and support of the paper! We will include the new experimental results into the next version manuscript.
> > >
> > > Best regards,
> > > Authors of 6000

---

### Official Review · Reviewer_J9bc · 2023-07-07

**Soundness:** 3 good
**Presentation:** 2 fair
**Contribution:** 3 good
**Rating:** 5
**Confidence:** 4

**Summary:**

This paper presents a method for controlling the output of a diffusion
model with multiple modalities of reference images, e.g. edges,
segmentation, depth, etc. It can be seen as proposing a multi-task
version of ControlNet. Experiments in the paper show the multi-task
approach outperforms the single-task, and also allows zero-shot
applicability to novel tasks such as colorization.

**Strengths:**

- The paper effectively demonstrates architecture modifications for multi-task controlling model
  and shows that it is generally better than the single-task (Table
  1).
- The model generalization to tasks that were not in the training set such
  as colorization, deblurring or inpainting is quite remarkable.
- The paper introduces a dataset with 20M multi-modal condition
  training pairs.

**Weaknesses:**

The paper presents a well engineered solution to achieving a
multi-task version of ControlNet, and show generalization to some new
tasks. Some decision justifications are not backed up by ablation
experiments and the experiments on task generalization are
demonstrated only with a few visual examples.

- If I understood correctly, there "Mixture of Experts" component is
  manually selecting the encoder for each modality. In that case
  calling this module a MOE is justified, since it could be misleading.
- Unless I missed it, the different components in the design are not
  ablated. What is the contribution of the Hypernet task embedding?
- How do you explain the poor performance of ControlNEt in
  Normal-to-Image (Fig. 6)
- Quantitative comparison for the generalization to new tasks is
  limited to a few visual examples. Some of the examples are even
  repeated three times (segmentation+skeleton); for that case, it may
  be more convincing if the paper showed three different results.

**Questions:**

Please refer to the weaknesses section.

**Limitations:**

Yes.

---

> ### Author Rebuttal · Authors · 2023-08-10
>
> We sincerely appreciate your insightful comments and suggestions for our submission.
>
> **Q1:  Missing Ablation Study**
>
> Thank you for pointing this out. We've conducted an ablation study with FID scores as follows. Our experimental setup adheres to Sec. 4.3 of the main paper, employing DDIM as the sampler and a guidance score of 9. Drawing from a collection of over 2,000 test samples sourced from Laion and COCO, we applied six tasks for evaluation, spanning edges (Canny, HED), regions (Seg), skeletons (Pose), and geometric maps (Depth, Normal). It's evident that the UniControl model (incorporating both MoE-Style Adapter and TaskHyperNet) consistently surpasses the results of its ablations.
>
> | MoE-Adapter | TaskHyperNet | Canny | HED | Depth | Normal  | Seg  |  Pose | Avg|
> |--|--|--|--|--|--|--|--|--|
> | x | x | 27.2 | 29.0 | 27.6 | 28.8 |29.1  | 30.2 | 28.7 |
> | ✓ | x | 24.5 | 26.1 | 23.7 | 24.8 | 26.9 | 28.3 | 25.7 |
> | ✓ | ✓ | **22.9**  | **23.6** | **21.3** | **23.4** | **25.5** | **27.4** | **24.0** |
>
> **Q2: UniControl is an engineered solution for multi-task version of ControlNet**
>
> UniControl is deeply influenced by the principles of HyperNetwork [1] and multi-task visual learning as Taskonomy [2]. It embodies the concept of "Control over the Control" or "Meta Control." We believe that unifying diverse visual modalities and tasks within a single framework is not just an engineering endeavor, but a significant scientific challenge since it is required to keep both single-task discriminability and multi-task generality.
>
> [1] HyperNetworks. ICLR 17.
>
> [2] Taskonomy: Disentangling Task Transfer Learning. CVPR 18.
>
>
>
>
> **Q3: Justification of MoE**
>
> Yes. It is not the MoE as its assembly is not learnable. Because of this distinction, we name it MoE-style adapter instead of MoE in the paper. But the high-level ideas parallel since each module catering to a specific modality, effectively functioning as an "Expert" network. When dealing with zero-shot tasks, the adapter's weights can be autonomously determined through the computation of similarity scores derived from task embeddings.
>
> The disparity in results can be attributed to the differing training data used for ControlNet and UniControl. Since ControlNet hasn't made its training data public, we were compelled to create our own datasets for both training and testing. In our approach, we selected images from Laion with a high resolution (>=512) and aesthetic scores exceeding 6. It's plausible that the quality of these images surpasses that of the original ControlNet’s training data for such tasks.
>
> **Q4: Question of Fig. 6 about the poor result of Normal-to-Image ControlNet**
>
> This Can be attributed to the differing training data used for ControlNet and UniControl. Since ControlNet hasn't made its training data public, we were compelled to create our own datasets for both training and testing. In our approach, we selected images from Laion with a high resolution (>=512) and aesthetic scores exceeding 6.  It's plausible that the quality of these images surpasses that of the original ControlNet’s training data for these tasks.
>
>
> **Q5: More results on zero-shot tasks**
>
> Thank you for your suggestion.  We've incorporated additional zero-shot results in the Appendix. It's worth noting that the zero-shot capabilities of UniControl are a welcome byproduct. An exhaustive exploration of arbitrary zero-shot success deserves its own comprehensive study and extends beyond the scope of our current focus. We remain hopeful that future research in this direction will exhibit the reliable zero-shot generalization capacities akin to those observed in LLMs.

---

> > ### Comment · Reviewer_J9bc · 2023-08-22
> > **Thank you for your rebuttal.**
> >
> > I thank the authors for the rebuttal and the new ablation studies showing the contribution of the Hypernet.
> > After reading the other reviewers' comments and authors' response, I am raising my score by one point.

---

> > > ### Author Response · Authors · 2023-08-22
> > > **After Feedback**
> > >
> > > Dear Reviewer J9bc,
> > >
> > > We sincerely thank you for your helpful comments and we are happy to hear that we've addressed most of the concerns. The new experimental results will be integrated to the next version manuscript.
> > >
> > > Best regards,
> > > Authors of 6000

---

### Official Review · Reviewer_hjKo · 2023-07-10

**Soundness:** 3 good
**Presentation:** 3 good
**Contribution:** 3 good
**Rating:** 6
**Confidence:** 4

**Summary:**

The paper presents UniControl, that unifies multiple visual controlling condition into a single unified model. To achieve this, the authors introduce a task-aware HyperNet to modulate the diffusion models, enabling adaptation to different condition-to-image (C2I) tasks simultaneously. UniControl is trained on nine unique C2I tasks and demonstrates impressive controlled generation quality. It also demonstrates zero-shot generation abilities with unseen visual conditions, including condition combination and new conditioning. User studies show that UniControl often surpasses the performance of single-task-controlled methods of comparable model sizes.

**Strengths:**

- The paper addresses an interesting and important research topic, unifying the controllability of different models into a single model.
- The authors compare the results with ControlNet and demonstrate better qualitative results.
- The zero-shot task generalization and instruction combination aspects of the proposed method are intriguing and valuable.

**Weaknesses:**

- The paper lacks quantitative evaluations for the alignment of the generated content and conditional inputs. For segmentation and bounding box, pretrained detectors could be used for evaluating the alignment, following the configuration in [1]. Although ControlNet does not handle bounding box conditions, a baseline for object detection instruction-following would be [1].

[1] Li, Y., Liu, H., Wu, Q., Mu, F., Yang, J., Gao, J., ... & Lee, Y. J. (2023). Gligen: Open-set grounded text-to-image generation. In Proceedings of the IEEE/CVF Conference on Computer Vision and Pattern Recognition (pp. 22511-22521).

- Zero-shot task generalization seems underexplored, and there is a lack of detailed analysis.
    - Inpainting results are not surprising as they share a strong alignment in terms of task instruction with the pretraining with outpainting.
    - For deblurring and colorization, it would be better to illustrate how the model can perform zero-shot task generalization. Does the capability mainly come from the task instruction encoder?
    - If so, it would be more intriguing to demonstrate results with more tasks that can really benefit from the zero-shot task generalization. It is easy to obtain data for colorization, inpainting. It would be valuable to demonstrate results on tasks that may be hard to collect training data.  For example, for scribble to image, ControlNet utilizes strong human-crafted data augmentation to synthesize the scribbles. Can UniControl generalize well to this case?

**Questions:**

- Can the authors please clarify if the difference between Ours-Single and ControlNet is only the training data / training schedule or there are other differences? From the results in Fig. 6 and Fig. 7, it seems that Ours is much better than ControlNet, while Ours is only slightly better than Ours-Single.

- Where does the performance can mainly come from ControlNet to Ours-Single? Is it because of the better dataset and/or longer training? Note that this is not a criticism, so I put it here in the questions. But I think a clarification or analysis in this regard can be helpful for the readers and the research community.

- Cost comparison with ControlNet. In L38-39, the authors mention "Retraining a separate model is necessary to handle a different modality of visual conditions, incurring non-trivial time and spatial complexity costs." However, UniControl requires 5000 GPU hours on A100, while ControlNet training for a single model is usually just 100-300 hours. This somehow invalidates this point.

**Limitations:**

The authors discuss the limitations in the main papar.

---

> ### Author Rebuttal · Authors · 2023-08-10
>
> Thanks for your valuable time to review this paper. We sincerely appreciate your constructive comments. Your concerns are addressed below.
>
> **Q1:  More quantitative evaluations for the alignment of the generated content and conditional inputs**
>
> Thank you for highlighting this. We agreed with your feedback and have incorporated additional quantitative evaluations. We've included FID metrics and the average precision (AP) metrics, adopting the methodology from GLIGEN [1], and utilizing pre-trained detectors.
>
> In addition to GLIGEN [1] as you suggested, we've also incorporated T2I-adapter [2] and ControlNet [3] as baselines, given their foundational contributions to this domain. Our experimental setup adheres to Section 4.3 of the main paper, and for sampling, we employ DDIM with a guidance score of 9. We've collected over 2,000 test samples from Laion and COCO, and conducted evaluations across six diverse tasks: these span edges (Canny, HED), regions (Seg), skeletons (Pose), and geometric maps (Depth, Normal).
>
> Our FID table, presented subsequently, showcases evaluations related to visual quality. A clear observation from the table is that our UniControl model consistently surpasses the baseline methods across the majority of tasks. Notably, UniControl also offers a more dense architecture compared to other single-task methods, which typically needs multiple checkpoints.
>
> FID Scores
>
> |  | GLIGEN | T2I-adapter  | ControlNet | UniControl |
> |--|--|--|--|--|
> | Canny | 24.9 | 23.6 | **22.7** | 22.9 |
> | HED | 27.8 | - | 25.1 | **23.6**  |
> | Depth | 25.8 | 25.4 | 25.5 | **21.3** |
> | Normal | 27.7 | - | 28.4 | **23.4** |
> | Seg | - | 27.1 | 26.7 | **25.5** |
> | Pose | - | 28.9 | 28.8 | **27.4** |
>
> Taking your advice, we've integrated the pre-trained YOLO-v4 detector. We deployed our UniControl model to generate images corresponding to the bounding box masks and captions from the COCO14-Val dataset. The summarized AP scores in the ensuing table demonstrate that our method is superior over GLIGEN.
>
> AP Scores on COCO (YOLO-V4)
>
> |  | GLIGEN | UniControl |
> |--|--|--|
> | AP | 24.0 | **26.2** |
> | AP_50 | 42.2 | **45.0** |
> | AP_75 | 24.1 | **26.3**  |
>
> [1] GLIGEN: Open-Set Grounded Text-to-Image Generation. CVPR 23.
>
> [2] T2I-Adapter: Learning Adapters to Dig out More Controllable Ability for Text-to-Image Diffusion Models. arXiv 2302.08453.
>
> [3] Adding Conditional Control to Text-to-Image Diffusion Models. arXiv 2302.05543.
>
> **Q2:  More details and exploration of the zero-shot tasks**
>
> *Inpainting and Outpainting:* While inpainting and outpainting might appear related, they are fundamentally distinct. Inpainting heavily leverages the contextual information from unmasked regions, necessitating a precise match. Conversely, outpainting has more freedom, with the generative model prioritizing prompts to envision new content. Directly using outpainting model for inpainting tasks can be challenging since the model tends to leave a sharp change over the mask boundaries. Our pretrained UniControl, thanks to intensive training across multiple tasks, has learned edge and region-to-image mappings, which assists in preserving contextual information. We've incorporated visual comparisons in the attached pdf for further clarity.
>
> *Deblurring and Colorization:* UniControl processes blurred or grayscale images as visual cues, relying on textual prompts and specific task instructions like "grey to image" or "image deblurring" during inference. The MoE-style adapters are structured based on the similarities observed among the pre-trained tasks, as explained in Sec. 3.3. For instance, the colorization task uses weights as "depth: 0.6, seg: 0.3, canny: 0.1". This adaptability stems from a blend of task-specific instructions and the MoE-style adapters. Misaligned adapters or inappropriate task instructions compromise the model's performance.
>
>
> *Scribbles:* Thank you for your suggestion. Indeed, our model demonstrates a promising capacity to generalize under scribble conditions, showing parallels to the ControlNet's ability, even though UniControl hasn't been directly trained using scribble data. Our pdf file provides additional results illustrating the scribble-to-image generation.
>
> **Q3:  Ours-Single vs ControlNet**
>
> The only difference between Ours-Single and ControlNet is the training data. The authors of ControlNet have not released their training data. As a result, it is necessary to reimplement the ControlNet model using our collected dataset, MultiGen-20M. Notably, MultiGen-20M is set to be the pioneering open-sourced conditional visual generation dataset in this area. The variance observed between Fig. 6 and Fig. 7 results from these differing datasets. We have filtered images with higher resolutions (>=512) and aesthetic scores (>6) from Laion whose quality would likely be better than the original ControlNet’s training data. The data quality is essential for visual performance. Therefore, Ours-single is better than the official ControlNet on these tasks.
>
> **Q4:  Fair comparison of training cost**
>
> Thank you for your careful observation.  Indeed, the ~5000 GPU Hours we've stated are similar to the cumulative training cost of multiple ControlNets. For our work, we utilized the A100-40G, whereas ControlNets employed the A100-80G. Let's break down ControlNets' training costs:
> |  | Canny | HED | Pose | Seg| Depth| Normal|Sketch | Total|
> |--|--|--|--|--|--|--|--|--|
> | Hours |600 | 300 | 300 | 400 | 500 |  200| 150 | 2450 |
> | GPU Type |A100-80G  | A100-80G | A100-80G | 3090TI | A100-80G |A100-80G  | A100-80G |   |
>
> As we can see, most ControlNets are trained by A100-80G whose total training cost is ~2500 GPU Hours.  In contrast, our use of the A100-40G, which has been shown to be less powerful than the A100-80G as per [4].
>
> [4] https://www.topcpu.net/en/gpu-c/a100-sxm4-40-gb-vs-a100-pcie-80-gb

---

> > ### Comment · Reviewer_hjKo · 2023-08-18
> >
> > I thank the authors for the detailed response. Most of my concerns are addressed by the authors' response, and I thus maintain my initial rating: weak accept. Please add these quantitative results in the revised paper. Thanks.

---

> > > ### Author Response · Authors · 2023-08-18
> > > **After Response**
> > >
> > > Thank you again for your understanding and constructive feedback. We are pleased to hear that most of your concerns have been addressed. As suggested, we will certainly make revisions to the manuscript to add these quantitative comparisons.
> > >
> > > Best regards,
> > > Authors of 6000

---

### Author Rebuttal · Authors · 2023-08-10

Thanks to all the reviewers for your valuable time in reviewing this paper. We sincerely appreciate your constructive comments and questions to make this paper better. Below, we respond to each concern in the order.

---

### Decision · Program_Chairs · 2023-09-21

**Decision:**

Accept (poster)

**Comment:**

This paper aims to consolidate different control conditions within a single model, distinguishing itself from ControlNet's approach of training separate models for each condition. It aligns more with the category of system papers and integrates a combination of existing techniques, including the original ZeroConv design from ControlNet, hard-gate Mixture of Experts (MOE), and HyperNet.

During the initial review phase, reviewers raised several concerns about the paper, primarily centered on the need for more comprehensive evaluation, analysis, and comparisons. This led to a varied range of feedback. However, the rebuttal effectively addressed most of these concerns, resulting in some reviewers who initially leaned towards a borderline rejection modifying their scores to borderline acceptance.

In summary, this paper hovers on the borderline of acceptance. After discussion, Area Chair finally recommends accept.